# Enhancing Diversity in Text-to-Image Generation without Compromising Fidelity

**Jiazhi Li**[†]        *jiazhil@usc.edu*
*Ming Hsieh Department of Electrical and Computer Engineering*
*University of Southern California*

**Mi Zhou**[†]        *mzhou91@gatech.edu*
*School of Electrical and Computer Engineering*
*Georgia Institute of Technology*

**Mahyar Khayatkhoei**        *mkhayat@isi.edu*
*Information Sciences Institute*
*University of Southern California*

**Jingyu Shi**[†]        *shi537@purdue.edu*
*Elmore Family School of Electrical and Computer Engineering*
*Purdue University*

**Xiang Gao**[†]        *gao2@cs.stonybrook.edu*
*Department of Computer Science*
*Stony Brook University*

**Jiageng Zhu**        *jiagengz@usc.edu*
*Ming Hsieh Department of Electrical and Computer Engineering*
*University of Southern California*

**Hanchen Xie**        *hanchenx@usc.edu*
*Thomas Lord Department of Computer Science*
*University of Southern California*

**Xiyun Song**        *xsong@futurewei.com*
*Futurewei Technologies Inc.*

**Zongfang Lin**        *zlin1@futurewei.com*
*Futurewei Technologies Inc.*

**Heather Yu**        *hyu@futurewei.com*
*Futurewei Technologies Inc.*

**Jieyu Zhao**        *jieyuz@usc.edu*
*Thomas Lord Department of Computer Science*
*University of Southern California*

**Reviewed on OpenReview:** *https://openreview.net/forum?id=180S4tOpmx&referrer=%5BAuthor% 20Console%5D(%2Fgroup%3Fid%3DTMLR%2FAuthors%23your-submissions)*

---

[†]The work was done during the internship at Futurewei Technologies Inc.

## Abstract

Effective text-to-image generation must synthesize images that are both realistic in appearance (sample fidelity) and have sufficient variations (sample diversity). Diffusion models have achieved promising results in generating high-fidelity images based on textual prompts, and recently, several diversity-focused works have been proposed to improve their demographic diversity by enforcing the generation of samples from various demographic groups. However, another essential aspect of diversity, sample diversity—which enhances prompt reusability to generate creative samples that reflect real-world variability—has been largely overlooked. Specifically, how to generate images that have sufficient demographic and sample diversity while preserving sample fidelity remains an open problem because increasing diversity comes at the cost of reduced fidelity in existing works. To address this problem, we first propose a bimodal low-rank adaptation of pretrained diffusion models which decouples the text-to-image conditioning, and then propose a lightweight bimodal guidance method that introduces additional diversity to the generation process using reference images retrieved through a fairness strategy by separately controlling the strength of text and image conditioning. We conduct extensive experiments to demonstrate the effectiveness of our method in enhancing demographic diversity (Intersectional Diversity (Shrestha et al., 2024)) by $2.47\times$ and sample diversity (Recall (Kynkäänniemi et al., 2019)) by $1.45\times$ while preserving sample fidelity (Precision (Kynkäänniemi et al., 2019)) compared to the baseline diffusion model.

## 1 Introduction

Diffusion models have made significant progress in generating high-fidelity content across various applications (*e.g.*, Text-to-Image (T2I) Generation) (Guo and Chen, 2024; Ho et al., 2020; Rombach et al., 2022; Dhariwal and Nichol, 2021). Despite the advancements in achieving better control over these models to produce high-fidelity content, there remains a lack of sufficient control in generating both high-fidelity and diverse content, particularly in terms of *demographic diversity* and *sample diversity*. Demographic diversity aims to address societal biases and ensure fair representation across diverse demographic groups (Wan et al., 2024a; Xu et al., 2018; Friedrich et al., 2023; Li et al., 2024b) while sample diversity seeks to improve prompt reusability to avoid monotonous or overly similar outputs and ensure that generated samples capture the variability of real-world samples (Zhang and Schomaker, 2022; Miao et al., 2024; Sadat et al., 2023; Li et al., 2021; Xia et al., 2021; Liu et al., 2020; Wang et al., 2024). Recently, several diversity-focused works (Friedrich et al., 2023; Esposito et al., 2023; Li et al., 2023a; Zhang et al., 2023a; Bansal et al., 2022; Luccioni et al., 2023; Perera and Patel, 2023; Bianchi et al., 2023) have been proposed to improve demographic diversity. However, enhancing sample diversity in diffusion models, particularly improving both aspects of diversity, remains overlooked.

Insufficient sample diversity in diffusion models hinders their further advancements in producing more photorealistic large-scale images (Marwood et al., 2023; Zameshina et al., 2023; Rassin et al., 2024; Li et al., 2024a). As shown in Fig. 1 (left, first row), images generated by state-of-the-art (SOTA) diffusion model (Stable Diffusion 3.5-Large (Esser et al., 2024)) appear repetitive, formulaic, and monotonous, resembling typical stock photos (*e.g.*, images generated with the prompt "Photo of a doctor" always feature a white wall background and a similar pose). This sharply contrasts with real-world images, which exhibit greater variation and creativity (Kynkäänniemi et al., 2019; Naeem et al., 2020), thereby limiting their broader applicability. Furthermore, SOTA models (Betker et al., 2023; Podell et al., 2023) are limited in large-scale image generation, as when generating 50-200 images from the same prompt, they often produce images similar to those already generated images even with different random initialization seeds (Tang et al., 2024; Du et al., 2024). We quantitatively demonstrate that SOTA models and existing diversity-focused methods struggle to capture real-world sample variability in Tab. 1 and Fig. 4, and exhibit limited prompt reusability in Fig. 6.

Since sample diversity is often overlooked in the literature, we propose a method to enhance it while also considering demographic diversity. It is well established that diversity and fidelity exhibit a trade-off (Dhariwal and Nichol, 2021; Ho and Salimans, 2022; Brock, 2018; Kingma and Dhariwal, 2018), meaning that improving

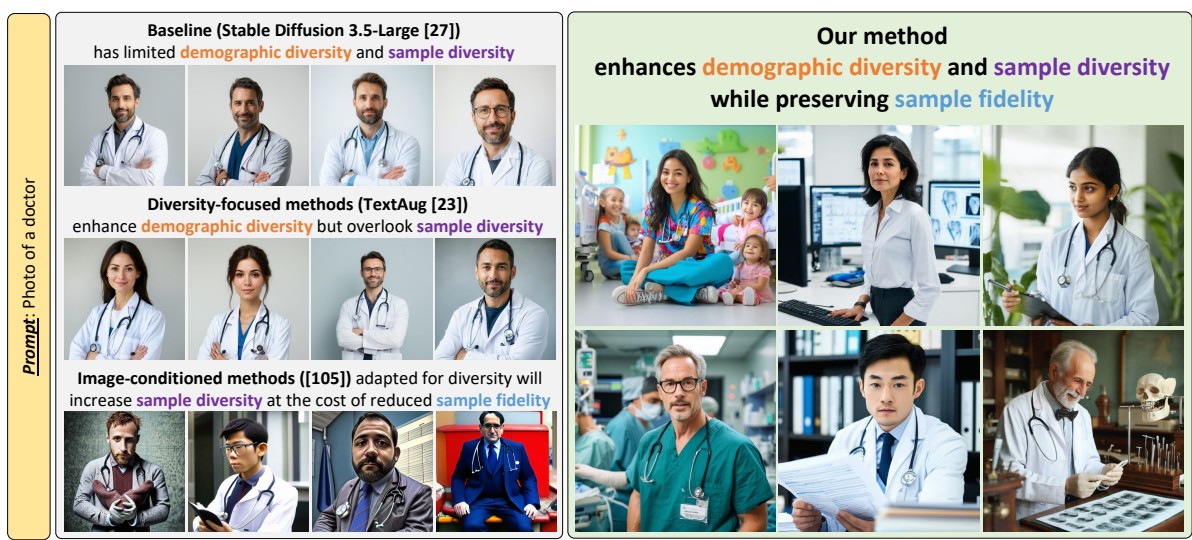

Figure 1: The proposed method (right) enhances both demographic diversity (*e.g.*, generating both female and male images) and sample diversity (*e.g.*, varied poses and backgrounds) while preserving sample fidelity (*e.g.*, sharp details, natural faces, and realistic lighting).

one often reduces the other. For instance, many general methods (Nichol et al., 2021; Blattmann et al., 2022), such as Classifier-Free Guidance (CFG) (Ho and Salimans, 2022), a widely adopted technique in diffusion models, trade off output variability over fidelity, as lower guidance scales introduce more details but reduce control. Consequently, a key challenge in improving diversity is effectively alleviating this diversity-fidelity trade-off, as both are essential for generating high-quality images. In this paper, our primary objective is to enhance both demographic and sample diversity while preserving sample fidelity.

To achieve our objective, we investigate whether *incorporating additional diversity can enhance output diversity without compromising fidelity*. To introduce more diversity, we alternate training between text and image modalities to encode image information and leverage the rich visual details from multiple retrieved reference images to augment the user prompt during inference. To preserve sample fidelity, we propose Bimodal Classifier-Free Guidance (BCFG), which extends the unified control of text and image modalities in CFG by separately controlling guidance from each modality. This technique allows image modality to fully exploit the added diversity while properly tuning text modality to maintain both sample fidelity and image-text alignment, as empirically verified in Fig. 8. Additionally, existing demographic diversity-focused methods (Shrestha et al., 2024) typically support only a fixed fairness criterion, which may cause overshooting biases (Wan and Chang, 2024). To address this limitation, we reformulate demographic diversity enhancement as a distribution alignment problem (Shen et al., 2024), enabling adaptability to various fairness criteria based on user needs. Finally, our method is implemented in a lightweight and efficient manner. Specifically, we design a low-rank joint text-image adapter architecture on the pre-trained model for bimodal conditioning. Our approach eliminates the computational overhead of re-training or Full Fine-Tuning (FFT) (Blattmann et al., 2022; Chen et al., 2022) by freezing all text modality parameters to preserve pre-trained knowledge and designing a delicate adapter for the image modality. We summarize our main contributions below:

- Proposing, to our knowledge, the first method to enhance both demographic diversity and sample diversity while preserving sample fidelity and alignment by alleviating the diversity-fidelity trade-off in a lightweight and efficient manner (12M parameters for Stable Diffusion v2.1).

- Highlighting and formulating sample diversity which is overlooked by existing diversity-focused methods.

- Introducing Low-Rank Image Adapter (LoRIA) for lightweight image-modality conditioning, Image-Augmented Prompt (IAP) for enhancing demographic and sample diversity under adaptable fairness criteria, and Bimodal Classifier-Free Guidance (BCFG) for alleviating the trade-off between diversity and fidelity.

- Providing an extensive empirical analysis of the proposed method, demonstrating its superior performance in enhancing demographic diversity (Intersectional Diversity (Shrestha et al., 2024)) from 0.19 to 0.47 and sample diversity (Recall) from 0.31 to 0.45 while maintaining sample fidelity (Precision) at 0.54, comparable to 0.52 of the baseline.

## 2 Related Work

**Enhancing Diversity in Diffusion Models.** While diffusion models can generate high-fidelity images (Ho et al., 2020; Rombach et al., 2022; Dhariwal and Nichol, 2021), they exhibit insufficient demographic (Wan et al., 2024a; He et al., 2024; Esposito et al., 2023) and sample diversity (Marwood et al., 2023; Cao et al., 2024; Naeem et al., 2020). To enhance demographic diversity, several methods directly augment the user prompt with demographic description (Bansal et al., 2022; Ding et al., 2021; Friedrich et al., 2023), while others apply Parameter-Efficient Fine-Tuning (Zhang et al., 2023a; Teo et al., 2024; Shen et al., 2024). Building on Retrieval-Augmented Generation (RAG) (Lewis et al., 2020; Cai et al., 2022), FairRAG (Shrestha et al., 2024) uses the user prompt (*e.g.*, "Photo of a doctor") to retrieve relevant images from external databases and boosts minority group sampling. Despite efforts to enhance demographic diversity (Luo et al., 2024; Chuang et al., 2023; Fraser et al., 2023; Gandikota et al., 2024), few methods improve sample diversity or both diversity in diffusion models.

**Diversity-Fidelity Trade-Off.** Diffusion models inherently exhibit the diversity-fidelity trade-off (Nichol et al., 2021; Blattmann et al., 2022), which limits high-quality data generation, as both aspects contribute to overall quality. Many general techniques (Dhariwal and Nichol, 2021; Ho and Salimans, 2022; Brack et al., 2023; Friedrich et al., 2023; Chen et al., 2022; Bansal et al., 2023; Epstein et al., 2023) enhance fidelity at the expense of diversity (Dhariwal and Nichol, 2021; Ho and Salimans, 2022; Brack et al., 2023; Friedrich et al., 2023; Chen et al., 2022; Bansal et al., 2023; Epstein et al., 2023). For instance, Classifier Guidance (CG) (Dhariwal and Nichol, 2021) increases fidelity by combining the original score estimate with the gradient from an auxiliary classifier to better align the sampling process with the conditioning information, albeit at the cost of reduced diversity. Classifier-Free Guidance (CFG) (Ho and Salimans, 2022) mathematically interprets the classifier gradient in CG as a combination of unconditional and conditional score estimates, removing the need for an explicit classifier. However, effectively enhancing diversity without compromising fidelity remains an open challenge.

## 3 Methods

We propose a lightweight and efficient method to enhance both demographic and sample diversity while preserving sample fidelity. Our method leverages the additional diversity introduced by retrieved reference images to alleviate the diversity-fidelity trade-off. During training, we design Low-Rank Image Adapter (LoRIA), a lightweight adapter architecture that extracts image information into a visual token and efficiently fuses text and image modalities to enable pre-trained models to incorporate reference images as additional conditioning. During inference, we introduce Image-Augmented Prompt (IAP), which retrieves reference images based on generated demographic descriptions to augment the original user prompt, thereby introducing additional sample diversity and enhancing demographic diversity in a manner adaptable to various fairness criteria. Furthermore, we propose Bimodal Classifier-Free Guidance (BCFG), which independently controls the guidance strength of text and image modalities, better aligning the sampling process with conditioning information from each modality.

### 3.1 Low-Rank Image Adapter (LoRIA)

To incorporate additional conditioning on reference images $i$, several methods (Blattmann et al., 2022; Chen et al., 2022; Ramesh et al., 2022) re-train or fine-tune pre-trained SD models which are originally

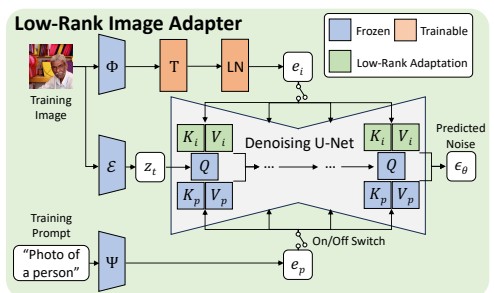 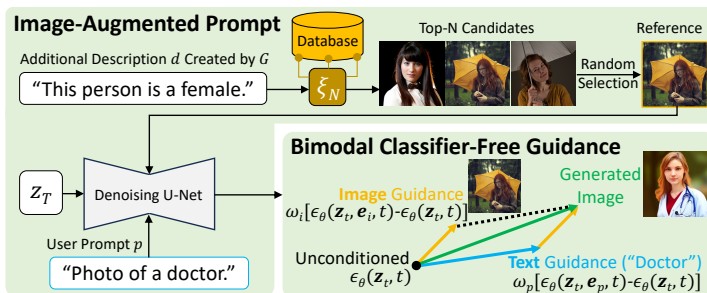

(a) During training, we alternate between text and image modalities to encode image information, and freeze all text modality parameters while training a linear projector T with Layer Normalization (LN) and applying LoRA for the image modality to incorporate additional image conditioning.

(b) During inference, we first retrieve Top-$N$ candidates using Diverse Retrieval Strategy $\xi$ based on specific fairness criteria, and then randomly select one or multiple images as reference images to construct Image-Augmented Prompt. Lastly, we use Bimodal Classifier-Free Guidance to leverage reference images to introduce additional diversity while applying stronger text guidance to preserve sample fidelity and text-image alignment.

Figure 2: The framework of the proposed Low-Rank Image Adapter, Image-Augmented Prompt, and Bimodal Classifier-Free Guidance.

conditioned solely on text prompts $p$. However, these methods are resource-intensive and time-consuming, and may compromise image-text alignment due to unrelated content in $i$ retrieved from limited external databases, as studied in Fig. 7. To address this and ensure that the image modality introduces variation without dominating generation, we freeze all text-modality parameters to retain pre-trained knowledge (Li et al., 2023e), while training a linear projector and applying Low-Rank Adaptation (LoRA)(Hu et al., 2021) to selectively update specific cross-attention weights in the modality fusion module (*e.g.*, U-Net(Ronneberger et al., 2015)), as illustrated in Fig. 2a. This lightweight design enables efficient image generation during both training and inference. Additionally, it helps prevent catastrophic forgetting (Kirkpatrick et al., 2017; Zhai et al., 2023; Smith et al., 2023) for the training prompt "Photo of a person", as analyzed in Appendix 4.2.

**Extracting Image Information into the Visual Token.** We utilize a frozen CLIP image encoder $\Phi$ followed by a trainable linear layer T and Layer Normalization (LN) (Ba, 2016) to obtain the visual token embedding $\mathbf{e}_i$. To ensure the model effectively extracts image information into the visual token, including demographic information, we train it on cases where only reference images are provided without any text prompt (Algorithm 1), which forces the model to rely maximally on the image for noise removal. As evidence, our ablation study (Tab. 3) demonstrates that omitting model training on cases with only image modality leads to a notable reduction in demographic diversity. Thus, during inference, the visual token is expected to retain demographic information from the retrieved reference images.

**Fusing Text and Image Modalities.** The bimodal embeddings $(\mathbf{e}_p, \mathbf{e}_i)$ are fed into SD for conditioning, with the text token embedding $\mathbf{e}_p$ obtained from a frozen CLIP text encoder $\Psi$. To fuse two modalities while considering their distinct roles, we introduce a *low-rank decoupled cross-attention layer*. The query features, shared across both modalities, are computed as $\mathbf{Q} = \mathbf{z}\mathbf{W}^q$, where $\mathbf{W}^q$ is the corresponding weight matrix. For text modality, the key and value features are obtained as $\mathbf{K}_p = \mathbf{e}_p\mathbf{W}_p^k$ and $\mathbf{V}_p = \mathbf{e}_p\mathbf{W}_p^v$, respectively. For image modality, the key and value features are calculated using weight matrices derived from the text weight matrices via LoRA, *i.e.*, $\mathbf{W}_i^k = \mathbf{W}_p^k + \mathbf{B}^k\mathbf{A}^k$ and $\mathbf{W}_i^v = \mathbf{W}_p^v + \mathbf{B}^v\mathbf{A}^v$, where $\mathbf{B}^k \in \mathbb{R}^{m \times r}$, $\mathbf{A}^k \in \mathbb{R}^{r \times n}$, $\mathbf{B}^v$, and $\mathbf{A}^v$ are low-rank matrices that approximate fine-tuning adjustments, with rank $r \ll \min(m, n)$. Overall, the layer output is formulated as:

$$\mathbf{z}' = \text{Attention}(\mathbf{Q}, \mathbf{K}_p, \mathbf{V}_p) + \text{Attention}(\mathbf{Q}, \mathbf{K}_i, \mathbf{V}_i), \tag{1}$$

where $\mathbf{Q} = \mathbf{z}\mathbf{W}^q$, $\mathbf{K}_p = \mathbf{e}_p\mathbf{W}_p^k$, $\mathbf{V}_p = \mathbf{e}_p\mathbf{W}_p^v$, $\mathbf{K}_i = \mathbf{e}_i(\mathbf{W}_p^k + \mathbf{B}^k\mathbf{A}^k)$, $\mathbf{V}_i = \mathbf{e}_i(\mathbf{W}_p^v + \mathbf{B}^v\mathbf{A}^v)$. During training, only $\mathbf{B}^k$, $\mathbf{A}^k$, $\mathbf{B}^v$, $\mathbf{A}^v$, and the linear layer T with LN are learnable while all other parameters remain frozen.

The training objective is to minimize the following loss function:

$$\mathcal{L} = \mathbb{E}_{\epsilon \sim \mathcal{N}(0,\mathbf{I}), \mathbf{z}_0, \mathbf{e}_p, \mathbf{e}_i, t \sim [1,T]} \left[ \| \epsilon - \epsilon_\theta(\mathbf{z}_t, \mathbf{e}_p, \mathbf{e}_i, t) \|^2 \right], \tag{2}$$

where $\epsilon$ is the sampled noise during the diffusion process; $\mathbf{z}_0 = \mathcal{E}(i)$ is the original latent representation of the training image $i$ encoded by a variational autoencoder (VAE) (Kingma, 2013) $\mathcal{E}$; $\mathbf{z}_t = \alpha_t \mathbf{z}_0 + \sigma_t \epsilon$ is the noisy version of $\mathbf{z}_0$ at timestep $t$, with $\alpha_t$ and $\sigma_t$ defining the diffusion process (Rombach et al., 2022); $\epsilon_\theta$ is the noise predicted by the denoising diffusion model parameterized by $\theta$; and $p$ is the fixed training prompt (*e.g.*, "Photo of a person"). To enhance the model's ability to extract information from each modality, we randomly discard conditioning during training. Specifically, we replace the training prompt with an empty sequence with probability $\pi_p = 0.1$, following (Saharia et al., 2022), and replace the image embedding with the all-zero embeddings of the same size with probability $\pi_i = 0.1$. The training and inference algorithms are shown in Algorithm 1 and Algorithm 2. We present the implementation details in Appendix 8.

---

**Algorithm 1** Training a Diffusion Model with Bimodal Classifier-Free Guidance

---

**Require:** $\pi_p$: the probability of training without text conditioning
**Require:** $\pi_i$: the probability of training without image conditioning
1: **repeat**
2:      $(\mathbf{z}_0, p, i) \sim \mathbf{p}(\mathbf{z}_0, p, i)$    ▷ Sample data $\mathbf{z}_0$, training prompt $p$, and reference images $i$ from the dataset
3:      $p \leftarrow \emptyset$ with probability $\pi_p$                       ▷ Discard text conditioning with probability $\pi_p$
4:      $\mathbf{e}_i \leftarrow \emptyset$ with probability $\pi_i$                    ▷ Discard image conditioning with probability $\pi_i$
5:      $t \sim [1, T]$                                    ▷ Sample a timestep uniformly from the range
6:      $\epsilon \sim \mathcal{N}(0, \mathbf{I})$                                      ▷ Generate Gaussian noise
7:      $\mathbf{z}_t = \alpha_t \mathbf{z}_0 + \sigma_t \epsilon$                    ▷ Corrupt the latent representation by adding noise
8:      Take a gradient step on $\nabla_\theta \left[ \| \epsilon - \epsilon_\theta(\mathbf{z}_t, \mathbf{e}_p, \mathbf{e}_i, t) \|^2 \right]$       ▷ Optimize the denoising model
9: **until** converged

---

**Comparison of Design Aspects between LoRIA and Existing Image Adapters (Ramesh et al., 2022; Ye et al., 2023; Zhang et al., 2023b; Mou et al., 2024).** The main design difference between existing methods and ours stems from different objectives: while they aim to edit the input image based on the user prompt (where the input image acts as the primary subject and its details need to be maximally preserved) (Ye et al., 2023), our method intends to leverage the image to introduce additional variation, hence enhancing diversity in T2I generation (where the input image serves as supportive context (Zhou et al., 2024)). As compared in Fig. 7, our method better aligns generated images with the user prompt by addressing potential unrelated content in reference images. To control the image modality strength, existing methods use an image scale parameter $\lambda$ (*e.g.*, scaling Attention($\mathbf{Q}, \mathbf{K}_i, \mathbf{V}_i$) in Eq. (1) (Ye et al., 2023)). However, adjusting this parameter during inference can introduce discrepancy between training and inference, which may induce flaws (*e.g.*, overexposure or lower resolution) (Lin et al., 2024b). Aligning it across training and inference requires retraining, which is time-consuming and limits flexibility for user control. To address this, we employ $\omega_i$ in BCFG, a training-free hyperparameter, to control the strength of image modality while preserving image quality, as shown in Fig. 3. We present the detailed comparison between control using the image scale parameter $\lambda$ and control via BCFG in Appendix 3.

## 3.2 Image-Augmented Prompt (IAP)

To introduce demographic diversity, conventional methods such as Text-Augmented Prompt (TAP) (Bianchi et al., 2023; Bansal et al., 2022; Ding et al., 2021) augment user prompt $p$ (*e.g.*, "Photo of a doctor") by appending it with fixed demographic descriptions $d$ (*e.g.*, "This person is a female."). However, TAP still exhibits limited sample diversity and struggles to substantially enhance demographic diversity due to linguistic ambiguity (*e.g.*, many *skin tone* categories are difficult to express using text alone) (Zhang et al., 2023a; Wan et al., 2024a; Shrestha et al., 2024). To address these limitations, we propose Image-Augmented Prompt (IAP), which enriches $p$ with the visual token encoded from one or multiple reference images $i$ retrieved based on the textual demographic description $d$. Incorporating the visual token not only conveys

---

**Algorithm 2** Inference with Bimodal Classifier-Free Guidance

---

**Require:** $p$: user prompt
**Require:** $i$: reference images
**Require:** $w_p$: text guidance scale
**Require:** $w_i$: image guidance scale
1: $\mathbf{z}_T \sim \mathcal{N}(0, \mathbf{I})$                        ▷ Initialize latent representation with Gaussian noise
2: **for** $t = T, \ldots, 1$ **do**
3:     $\tilde{\epsilon}_\theta(\mathbf{z}_t, \mathbf{e}_p, \mathbf{e}_i, t) = \epsilon_\theta(\mathbf{z}_t, t) + \omega_p[\epsilon_\theta(\mathbf{z}_t, \mathbf{e}_p, t) - \epsilon_\theta(\mathbf{z}_t, t)] + \omega_i[\epsilon_\theta(\mathbf{z}_t, \mathbf{e}_i, t) - \epsilon_\theta(\mathbf{z}_t, t)]$    ▷ Compute BCFG score
4:     $\mathbf{z}_{t-1} = (\mathbf{z}_t - \sigma_t \tilde{\epsilon}_\theta(\mathbf{z}_t, \mathbf{e}_p, \mathbf{e}_i, t))/\alpha_t$             ▷ Update latent representation for the next timestep
5: **end for**
6: **return** $\mathbf{z}_0$                          ▷ Return the final denoised latent representation

---

demographic information to enhance demographic diversity but also introduces additional sample diversity through rich visual details (*e.g.*, appearance, environment, and lighting conditions) that text alone cannot provide, thereby increasing sample diversity. We mathematically prove (Theorem 1 in Appendix 1.1) and empirically verify (Tab. 1) that IAP generates more diverse images than TAP.

**Diverse Retrieval Strategy (DRS).** To further improve diversity, we introduce DRS, a two-step retrieval strategy $\xi$, as shown in Fig. 2b. First, we perform nearest neighbor search (Borgeaud et al., 2022) with demographic information $d$ to retrieve Top-$N$ candidates (Blattmann et al., 2022) based on cosine similarity between their CLIP-encoded text embeddings $\mathbf{e}_d$ and image embeddings $\mathbf{e}_i$ (Radford et al., 2021). Next, we randomly select one or more samples as reference images $i$ for each generation query $q$. This random selection contributes to generating diverse samples since various image candidates representing $d$, rather than a fixed text description $d$, can reduce biases introduced by deterministic selection. Compared with TAP $(p, d)$, IAP is defined as $(p, i)$, where $i = \xi(d)$. With IAP $(p, i)$ integrating both text $p$ and image $i$, we explore a method to separately control the strength of each modality in Sec. 3.3.

**Adaptable Description Generator (ADG).** Existing demographic diversity-focused methods (Bansal et al., 2022; Friedrich et al., 2023; Zhang et al., 2023a; Teo et al., 2024; Shen et al., 2024; Shrestha et al., 2024) are designed for a specific fairness criterion and limit sufficient control over demographic distribution of generated images, which may cause overshooting biases (Wan and Chang, 2024; Wan et al., 2024b) since perceptions of fairness vary across contexts. To address this, we formulate demographic diversity enhancement as a distributional alignment problem (Shen et al., 2024) and propose ADG, which generates $d$ for image retrieval based on a target distribution $\mathcal{D}$ specified by user-defined fairness criteria, making it adaptable to various fairness criteria. For a generation query $q$ with prompt $p$, we use the target distribution $\mathcal{D}_p(a_1, ..., a_j)$ of demographic attributes $A_1, ..., A_j$ to direct a description generator $G(\mathcal{D}_p, \Lambda)$ in creating demographic description $d_q$ by filling a template $\Lambda$ (*e.g.*, "This person is a [AGE], [SKIN TONE] [SEX].") with respective demographic qualifiers[*]. Note that $\mathcal{D}_p$ and $\Lambda$ can be adapted to various fairness criteria such as disparate impact (Esposito et al., 2023), demographic factuality (Wan et al., 2024b), counter-stereotypes (Bianchi et al., 2023), or other user-defined fairness criteria; implementations are in Appendix 2. In the main paper, following (Friedrich et al., 2023; Xu et al., 2018; Shrestha et al., 2024), we consider a well-known fairness criterion, demographic parity (DP) (Hardt et al., 2016), to define $\mathcal{D}_p$. DP requires the independence between the label $Y$ (*e.g.*, occupation) and demographic attribute $A$ (*e.g.*, sex) for a (generated/synthetic) dataset, *i.e.*, $P(Y|A) = P(Y)$. Thus, to achieve DP and generate an overall balanced dataset, $\mathcal{D}_p$ is a uniform distribution[†] for any $p$. This DP-based augmentation forms the foundation for the proposed IAP.

### 3.3 Bimodal Classifier-Free Guidance (BCFG)

Classifier-Free Guidance (CFG) (Ho and Salimans, 2022) is commonly used in diffusion-based models (Ho et al., 2020; Rombach et al., 2022; Nichol et al., 2021) to direct the inverse diffusion process towards condi-

---

[*]We present the values for demographic qualifiers in Appendix 2.1.
[†]We present the proof in Appendix 2.2.

Figure 3: Visualization of varying image guidance scale $\omega_i$ from 1 to 10 in BCFG (Eq. (4)), with text guidance scale $\omega_p$ fixed at 7.5. $\omega_i$ controls how much the inverse diffusion process path will be pulled towards features of the reference image (*e.g.*, increasing $\omega_i$ shortens the doctor's sleeves to match the reference image). This allows users to easily choose their desired degree of diversity by varying $\omega_i$.

tioning features by incorporating the predicted noise during sampling:

$$\tilde{\epsilon}_\theta(\mathbf{z}_t, \mathbf{e}_p, t) = \epsilon_\theta(\mathbf{z}_t, t) + \omega[\epsilon_\theta(\mathbf{z}_t, \mathbf{e}_p, t) - \epsilon_\theta(\mathbf{z}_t, t)], \tag{3}$$

where $\epsilon_\theta(\mathbf{z}_t, \mathbf{e}_p, t)$ and $\epsilon_\theta(\mathbf{z}_t, t)$ are the conditional and unconditional score estimate, respectively; $\omega > 0$ is the guidance scale; $\mathbf{z}_t$ is the intermediate representation of the U-Net (Ronneberger et al., 2015) at timestep $t$; and $\mathbf{e}_p$ is the text embedding of user prompt $p$. However, CFG is well known for exhibiting the diversity-fidelity trade-off, where reducing $\omega$ enhances diversity at the cost of fidelity (Dhariwal and Nichol, 2021; Ho and Salimans, 2022; Nichol et al., 2021). In this work, we propose two modifications to CFG that alleviate this trade-off, thereby enabling diffusion models to achieve greater diversity while preserving sample fidelity.

First, we extend CFG to incorporate reference images $i$ (introduced by IAP) by replacing $\epsilon_\theta(\mathbf{z}_t, \mathbf{e}_p, t)$ with $\epsilon_\theta(\mathbf{z}_t, \mathbf{e}_p, \mathbf{e}_i, t)$ in Eq. (3) where $\mathbf{e}_i$ is the image embedding of $i$. By introducing image modality, rich visual details and variations from different reference images can introduce additional diversity beyond the textual description $d$, even though the user prompt $p$ remains unchanged.

Second, to ensure that incorporating diversity does not compromise sample fidelity, we extend the unified control of text and image modalities in CFG by separately controlling guidance from two modalities. As empirically demonstrated in Fig. 8, using a unified guidance scale $\omega$ for both modalities leads to suboptimal sample fidelity and text-image alignment (Appendix 4.1) due to potentially conflicting content in $i$ (*e.g.*, a skier) that contradicts $p$ (*e.g.*, "Photo of a doctor") (Teo et al., 2024). However, in T2I generation, the generated image should primarily reflect user-provided prompt $p$ (Li et al., 2019; Qiao et al., 2019; Ding et al., 2021). Thus, in our proposed IAP $(p, i)$, the user prompt $p$ serves as the primary role, while reference images $i$ serve as the supportive role, introducing additional variations rather than imposing all details that could dominate generated images. To achieve this, we propose Bimodal Classifier-Free Guidance (BCFG), which enables separate control over the text prompt and reference image modalities by leveraging the predicted noise during sampling:

$$\begin{aligned}
\tilde{\epsilon}_\theta(\mathbf{z}_t, \mathbf{e}_p, \mathbf{e}_i, t) = \epsilon_\theta(\mathbf{z}_t, t) &+ \omega_p[\epsilon_\theta(\mathbf{z}_t, \mathbf{e}_p, t) - \epsilon_\theta(\mathbf{z}_t, t)] \\
&+ \omega_i[\epsilon_\theta(\mathbf{z}_t, \mathbf{e}_i, t) - \epsilon_\theta(\mathbf{z}_t, t)],
\end{aligned} \tag{4}$$

where $\omega_p > 0$ and $\omega_i > 0$ are guidance scales for user prompt $p$ and reference images $i$. We set $\omega_p$ as 7.5 following (Ho and Salimans, 2022) and choose $\omega_i < \omega_p$ to prioritize text modality in image generation. In Fig. 8, we empirically compare BCFG with CFG on the diversity-fidelity trade-off, showing that BCFG can enhance diversity while preserving fidelity. Besides, we compare it with other guidance methods (Ho and Salimans, 2022; Brack et al., 2023; Chen et al., 2022) and analyze $\omega_i$ at different scales in Appendix 4.1.

**The Corresponding Sampling Distribution of Bimodal Classifier-Free Guidance (BCFG).** As interpreted below, the predicted noise in BCFG yields approximate samples from the following distribution[‡],

$$\tilde{p}_\theta(\mathbf{z}_t | \mathbf{e}_p, \mathbf{e}_i) \propto p_\theta(\mathbf{z}_t) p_\theta(\mathbf{e}_p | \mathbf{z}_t)^{\omega_p} p_\theta(\mathbf{e}_i | \mathbf{z}_t)^{\omega_i}. \tag{5}$$

Intuitively, the effect of BCFG is to increase the sampling probability of data points with a higher likelihood of matching the user prompt $p$ and reference image $i$ by the implicit classifier (Ho and Salimans, 2022).

---

[‡]For brevity, we omit the timestep variable $t$.

In Sec. 4, we apply BCFG to the proposed bimodal conditioning module (LoRIA) to control the strength of each modality. Notably, BCFG can be applied independently of the conditioning module, as empirically verified with other conditioning modules in Sec. 4.6.

**Intuition of the Corresponding Sampling Distribution.** We now provide a mathematical interpretation of the sample distribution of Bimodal Classifier-Free Guidance. Recall that the diffusion score (Ho and Salimans, 2022)

$$\epsilon_\theta(\mathbf{z}_t, \mathbf{e}_p, \mathbf{e}_i) = -\sigma_t \nabla_{\mathbf{z}_t} \log p(\mathbf{z}_t, \mathbf{e}_p, \mathbf{e}_i) \tag{6}$$

$$= -\sigma_t \nabla_{\mathbf{z}_t} \log[p(\mathbf{e}_i, \mathbf{e}_p) p(\mathbf{z}_t | \mathbf{e}_i, \mathbf{e}_p)] \tag{7}$$

$$= -\sigma_t \nabla_{\mathbf{z}_t} [\log p(\mathbf{e}_i, \mathbf{e}_p) + \log p(\mathbf{z}_t | \mathbf{e}_i, \mathbf{e}_p)] \tag{8}$$

$$\approx -\sigma_t \nabla_{\mathbf{z}_t} \log p(\mathbf{z}_t | \mathbf{e}_i, \mathbf{e}_p) \tag{9}$$

since $p(\mathbf{e}_i, \mathbf{e}_p)$ is not a function of $\mathbf{z}_t$.

Suppose we have two auxiliary implicit classifier models $p_\theta(\mathbf{e}_p | \mathbf{z}_t) \propto \frac{p(\mathbf{z}_t | \mathbf{e}_p)}{p(\mathbf{z}_t)}$ and $p_\theta(\mathbf{e}_i | \mathbf{z}_t) \propto \frac{p(\mathbf{z}_t | \mathbf{e}_i)}{p(\mathbf{z}_t)}$. Assume exact estimate $\epsilon^*(\mathbf{z}_t, \mathbf{e}_p)$ of $p(\mathbf{z}_t | \mathbf{e}_p)$, $\epsilon^*(\mathbf{z}_t, \mathbf{e}_i)$ of $p(\mathbf{z}_t | \mathbf{e}_i)$, and $\epsilon^*(\mathbf{z}_t)$ of $p(\mathbf{z}_t)$. The gradient of the resulting classifier can be written as $\nabla_{\mathbf{z}_t} \log p(\mathbf{e}_p | \mathbf{z}_t) = -\frac{1}{\omega_p} [\epsilon^*(\mathbf{z}_t, \mathbf{e}_p) - \epsilon^*(\mathbf{z}_t)]$ and $\nabla_{\mathbf{z}_t} \log p(\mathbf{e}_i | \mathbf{z}_t) = -\frac{1}{\omega_i} [\epsilon^*(\mathbf{z}_t, \mathbf{e}_i) - \epsilon^*(\mathbf{z}_t)]$ respectively. Since the exact scores $\epsilon^*(\mathbf{z}_t, \mathbf{e}_p)$, $\epsilon^*(\mathbf{z}_t, \mathbf{e}_i)$, and $\epsilon^*(\mathbf{z}_t)$ are not available, we use their estimates $\epsilon_\theta(\mathbf{z}_t, \mathbf{e}_p)$, $\epsilon_\theta(\mathbf{z}_t, \mathbf{e}_i)$, and $\epsilon_\theta(\mathbf{z}_t)$ respectively. The modified score function in Eq. (4) can thus be rewritten as

$$\epsilon_\theta(\mathbf{z}_t, t) + \omega_p [\epsilon_\theta(\mathbf{z}_t, \mathbf{e}_p, t) - \epsilon_\theta(\mathbf{z}_t, t)] +$$
$$\omega_i \ [\epsilon_\theta(\mathbf{z}_t, \mathbf{e}_i, t) - \epsilon_\theta(\mathbf{z}_t, t)] \tag{10}$$

$$\approx \epsilon_\theta(\mathbf{z}_t, t) - \sigma_t \omega_p \nabla_{\mathbf{z}_t} \log p_\theta(\mathbf{e}_p | \mathbf{z}_t) -$$
$$\sigma_t \omega_i \nabla_{\mathbf{z}_t} \log p_\theta(\mathbf{e}_i | \mathbf{z}_t) \tag{11}$$

$$= -\sigma_t \nabla_{\mathbf{z}_t} [\log p_\theta(\mathbf{z}_t) + \omega_p \log p_\theta(\mathbf{e}_p | \mathbf{z}_t) + \omega_i \log p_\theta(\mathbf{e}_i | \mathbf{z}_t)] \tag{12}$$

$$= -\sigma_t \nabla_{\mathbf{z}_t} \log[p_\theta(\mathbf{z}_t) p_\theta(\mathbf{e}_p | \mathbf{z}_t)^{\omega_p} p_\theta(\mathbf{e}_i | \mathbf{z}_t)^{\omega_i}] \tag{13}$$

$$= -\sigma_t \nabla_{\mathbf{z}_t} \log[p_\theta(\mathbf{z}_t) p_\theta(\mathbf{e}_p | \mathbf{z}_t) p_\theta(\mathbf{e}_i | \mathbf{z}_t)$$
$$p_\theta(\mathbf{e}_p | \mathbf{z}_t)^{\omega_p - 1} p_\theta(\mathbf{e}_i | \mathbf{z}_t)^{\omega_i - 1}] \tag{14}$$

$$= -p(\mathbf{e}_p, \mathbf{e}_i) \sigma_t \nabla_{\mathbf{z}_t} [p_\theta(\mathbf{z}_t | \mathbf{e}_p, \mathbf{e}_i) p_\theta(\mathbf{e}_p | \mathbf{z}_t)^{\omega_p - 1} p_\theta(\mathbf{e}_i | \mathbf{z}_t)^{\omega_i - 1}] \tag{15}$$

where $p(\mathbf{e}_p, \mathbf{e}_i)$ is a positive constant. In the last step, we used the Bayes Rule and the independence of random variable $\mathbf{e}_p$ and $\mathbf{e}_i$ given $\mathbf{z}_t$ to derive $p(\mathbf{z}_t | \mathbf{e}_p, \mathbf{e}_i) = \frac{p(\mathbf{z}_t, \mathbf{e}_p, \mathbf{e}_i)}{p(\mathbf{e}_p, \mathbf{e}_i)} = \frac{p(\mathbf{e}_p, \mathbf{e}_i | \mathbf{z}_t) p(\mathbf{z}_t)}{p(\mathbf{e}_i, \mathbf{e}_p)} = \frac{p(\mathbf{z}_t) p(\mathbf{e}_p | \mathbf{z}_t) p(\mathbf{e}_i | \mathbf{z}_t)}{p(\mathbf{e}_p, \mathbf{e}_i)}$. This leads to Eq. (5).

## 4 Experimental Evaluation

### 4.1 Experiment Setup

**Datasets and Evaluation Prompts** Following (Shrestha et al., 2024), we construct non-overlapping training and reference image datasets from human images in MSCOCO (Lin et al., 2014) and OpenImages-v6 (Krasin et al., 2017), and store the pre-computed CLIP image embeddings (Radford et al., 2021) to speed up the inference process by bypassing the usage of the image encoder during inference. For evaluation, we use prompts of 80 occupations that exhibit bias toward specific demographic groups (Shrestha et al., 2024; Friedrich et al., 2023) and generate 10,000 images (*i.e.*, 125 per prompt). Specifically, we employ the template "Photo of a [OCCUPATION]" to create prompts such as "Photo of a doctor". Additional details are provided in Appendix 7.

**Metrics** For demographic diversity, we use intersectional diversity (Shrestha et al., 2024), calculated as the normalized entropy of unique demographic groups categorized by sex, age, and skin tone, and individual diversity (Shrestha et al., 2024) to measure each attribute separately. To obtain demographic labels, we follow prior work (Shrestha et al., 2024; Cho et al., 2023) and use a pre-trained classifier to generate pseudo labels.

Table 1: Comparison of our method with existing methods. **Bold** indicates the best results, and underline indicates the second-best results.

| Category | Method | Demographic Diversity ↑ | | | | Sample Diversity | | Sample Fidelity | | Sample Quality | | Alignment |
|---|---|---|---|---|---|---|---|---|---|---|---|---|
| | | Sex | Age | Skin Tone | Intersec. | W1KP ↓ | Recall ↑ | IS ↑ | Precision ↑ | FID ↓ | $F1_{PR}$ ↑ | CLIP ↑ |
| Baseline | SDv2.1 Rombach et al. (2022) | 0.27 | 0.22 | 0.22 | 0.19 | 0.88 | 0.31 | 22.81 | 0.52 | 27.87 | 0.39 | 23.63 |
| Existing Demographic Diversity Methods | Interven Bansal et al. (2022) | 0.45 | 0.44 | 0.36 | 0.33 | 0.85 | 0.29 | 19.93 | 0.46 | 32.71 | 0.36 | 23.19 |
| | TextAug Ding et al. (2021) | 0.77 | 0.43 | 0.33 | 0.34 | 0.80 | 0.28 | 16.03 | 0.49 | 30.81 | 0.36 | 23.02 |
| | FairDiff Friedrich et al. (2023) | 0.37 | 0.23 | 0.22 | 0.20 | 0.83 | 0.34 | 22.40 | 0.50 | 27.77 | 0.40 | **23.97** |
| | ITI-GEN Zhang et al. (2023a) | 0.80 | 0.56 | 0.38 | 0.44 | 0.70 | 0.34 | 20.56 | 0.51 | 34.02 | 0.41 | 21.44 |
| | FairQueue Teo et al. (2024) | 0.82 | 0.56 | 0.41 | 0.47 | 0.72 | 0.35 | 21.23 | 0.52 | 30.43 | 0.42 | 22.65 |
| | DAL Shen et al. (2024) | **0.83** | **0.57** | 0.42 | 0.43 | 0.79 | 0.33 | **23.50** | 0.51 | 28.77 | 0.40 | 23.12 |
| | FairRAG Shrestha et al. (2024) | 0.80 | 0.56 | **0.42** | 0.44 | 0.61 | 0.37 | 20.88 | 0.53 | 26.82 | 0.44 | 23.63 |
| Image-Conditioned Methods Adapted for Diversity Enhancement | RDM Blattmann et al. (2022) | 0.43 | 0.22 | 0.38 | 0.23 | 0.57 | 0.40 | 18.52 | 0.52 | 25.24 | 0.45 | 23.78 |
| | unCLIP Ramesh et al. (2022) | 0.50 | 0.27 | 0.26 | 0.23 | 0.72 | 0.44 | 18.09 | 0.47 | 41.33 | 0.45 | 21.22 |
| | IP-Adapter Ye et al. (2023) | 0.55 | 0.22 | 0.27 | 0.22 | 0.65 | **0.45** | 19.88 | 0.50 | 26.38 | 0.47 | 22.28 |
| | T2I-Adapter Mou et al. (2024) | 0.37 | 0.24 | 0.16 | 0.16 | 0.70 | 0.42 | 18.55 | 0.51 | 62.55 | 0.46 | 22.54 |
| | ControlNet Zhang et al. (2023b) | 0.43 | 0.29 | 0.18 | 0.20 | 0.65 | 0.43 | 18.49 | 0.49 | 58.25 | 0.46 | 22.60 |
| | Ours | 0.82 | **0.57** | **0.42** | **0.47** | **0.54** | **0.45** | 22.45 | **0.54** | **23.18** | **0.49** | 23.05 |

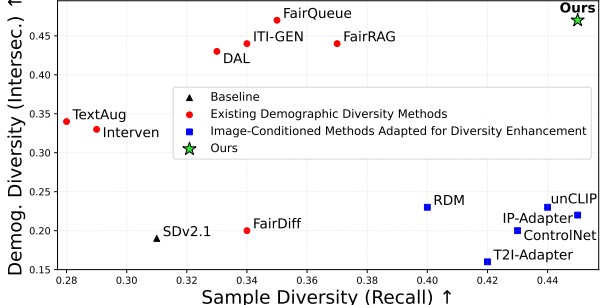

(a) Existing methods fail to enhance both demographic and sample diversity.

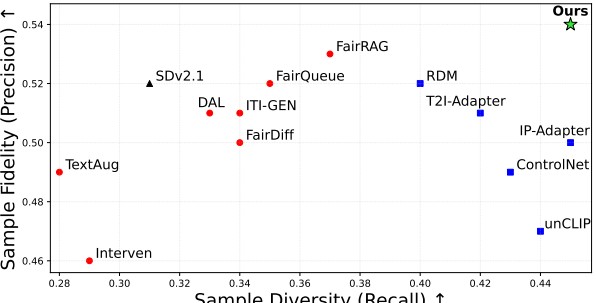

(b) Image-conditioned methods increase sample diversity but lack fidelity.

Figure 4: Our method (top-right) enhances both demographic diversity and sample diversity **simultaneously** while preserving sample fidelity, whereas existing methods can only improve either demographic diversity or sample diversity while sacrificing sample fidelity.

While this may introduce bias Li and Abd-Almageed (2021), we mitigate concerns through sanity checks comparing classifier outputs with manual annotations on a random subset, achieving >90% agreement. Still, reliance on external classifiers underscores the need for fairer, more robust evaluation tools in future work. For sample diversity, we use Recall (Kynkäänniemi et al., 2019) and W1KP (Tang et al., 2024). We use a fixed set of samples from MSCOCO that do not overlap with the reference image dataset as real samples to compute metrics involving real samples (Brock, 2018). Additionally, we use W1KP, a metric independent of real samples, to evaluate sample diversity and prompt reusability, as it is more effective than other diversity metrics (*e.g.*, LPIPS (Zhang et al., 2018) and ST-LPIPS (Ghildyal and Liu, 2022)) (Tang et al., 2024). We use Precision (Kynkäänniemi et al., 2019) and Inception Score (IS) (Salimans et al., 2016) to evaluate sample fidelity. We use FID (Heusel et al., 2017) to assess overall sample quality, as it captures both diversity and fidelity (Dhariwal and Nichol, 2021; Karras et al., 2019; 2020). Additionally, we calculate the harmonic mean of Precision and Recall, denoted as $F1_{PR}$, to comprehensively evaluate both sample diversity and sample fidelity. We use CLIP Score (Radford et al., 2021) to evaluate the alignment between the user prompt and the actual content of the generated image.

## 4.2 Comparison with Existing Demographic Diversity Methods

We compare our method with an extensive list of existing demographic diversity methods in Tab. 1 and Figs. 4 and 6. Since most compared methods are built on SDv2.1, we use it as the backbone in this comparison, and later evaluate our method on other backbones in Fig. 9, including the SOTA SDv3.5-L (Esser et al., 2024). In Tab. 1 and Fig. 5, our method outperforms all demographic diversity methods in improving sample

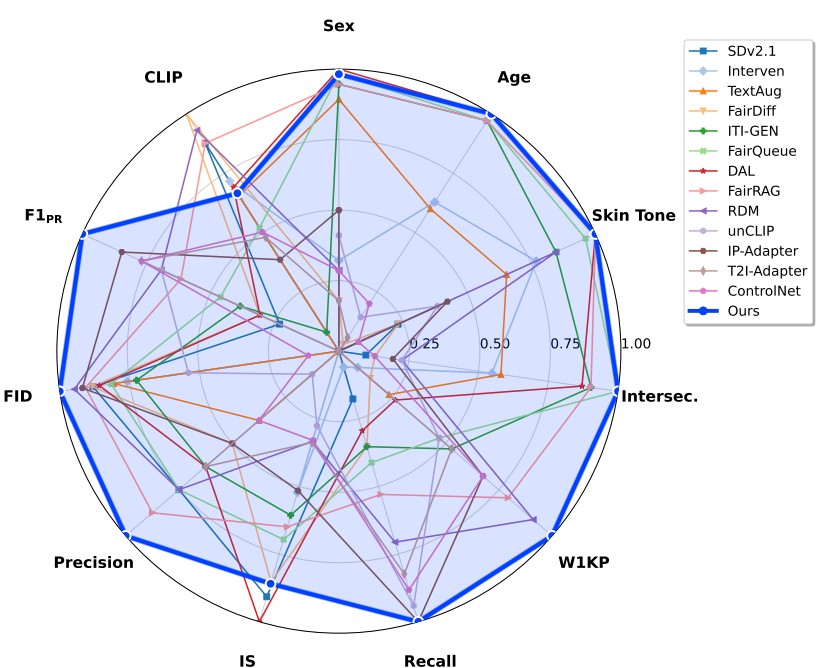

Figure 5: Radar plot aggregating multiple metrics from Tab. 1 into a single visualization. Scores are min–max normalized to the range $[0, 1]$ as $x_{\mathrm{norm}} = \frac{x - x_{\min}}{x_{\max} - x_{\min}}$, where $x_{\min}$ and $x_{\max}$ are the minimum and maximum values across methods. For metrics where lower is better, we apply an inversion $x_{\mathrm{inv}} = 1 - x_{\mathrm{norm}}$ to ensure consistent evaluation.

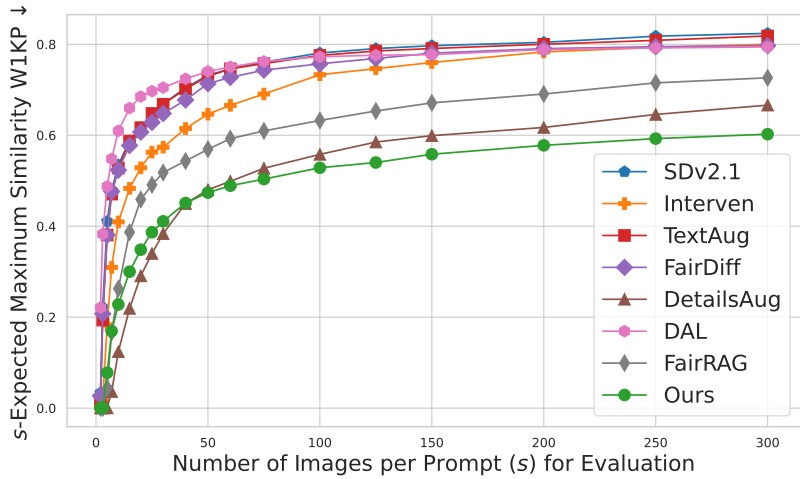

Figure 6: Prompt reusability evaluated by W1KP Tang et al. (2024) (lower is better). Our method performs the best compared to other methods.

diversity, while having on-par or better demographic diversity and sample fidelity. For instance, compared to FairRAG (Shrestha et al., 2024), which retrieves reference images using $p$, our method achieves high sample diversity by using description $d$, as more candidates match $d$ than $p$. In addition, beyond FairRAG which considers only one single criterion, we reframe demographic diversity enhancement as a distribution alignment problem and introduce an Adaptable Description Generator that supports multiple fairness criteria. We also propose Bimodal CFG (BCFG) to control text and image guidance separately, and design an improved image adaptor architecture that preserves fidelity while enhancing diversity. Interestingly,

our method also surpasses TAP-based methods (Bansal et al., 2022; Ding et al., 2021; Friedrich et al., 2023) in demographic diversity, particularly in skin tone, possibly because skin tone is difficult to specify in text but easily conveyed through images (Zhang et al., 2023a). Moreover, our method achieves comparable performance in sample fidelity to the baseline (*e.g.*, precision 0.54 *vs.* 0.52 and FID 23.18 *vs.* 27.87). We attribute the FID improvement to increased sample diversity, as FID captures both diversity and fidelity. In Fig. 6, where we evaluate prompt reusability across representative methods with greater sample diversity, our method performs the best.

## 4.3 Adapting Image-Conditioned Methods for Enhancing T2I Diversity

Despite a lack of existing solutions for enhancing sample diversity in T2I generation, it might be possible to construct such a solution by simply combining existing image-conditioned methods with a retrieval strategy. We implement such a combination as follows: for any given user prompt, we first retrieve reference images using a diverse pre-existing dataset and then provide the reference images as prompts to image-conditioned generative methods to synthesize the final image. We consider several SOTA image-conditioned methods, including RDM (Blattmann et al., 2022), unCLIP (Ramesh et al., 2022), IP-Adapter (Ye et al., 2023), T2I-Adapter (Mou et al., 2024) and ControlNet (Zhang et al., 2023b), under this pipeline and study their effectiveness in enhancing sample diversity while preserving sample fidelity. In Fig. 4 and Tab. 1, we observe that they can increase sample diversity, but it comes at the cost of reducing fidelity, whereas our proposed method enhances both demographic and sample diversity without compromising fidelity. To elucidate the source of their limitation, in Fig. 7 we show how these methods rely strongly on the relevance between user prompts and retrieved reference images, which can degrade sample quality in generating using large-scale diverse datasets due to limited relevant references. In contrast, our method uses BCFG to separately handle text and image modalities and prioritize text guidance for greater robustness to unrelated content in reference images.

## 4.4 Alleviating the Diversity-Fidelity Trade-Off

We study BCFG in alleviating the diversity/fidelity trade-off by comparing it with Classifier-Free Guidance (CFG) (Ho and Salimans, 2022) across guidance scales $\omega$ from 1 to 20. For BCFG, we set $\omega_p = 7.5$ following the setting in (Ho and Salimans, 2022) and vary $\omega_i$ from 1 to 20. For CFG, we use two settings: (1) CFG (*Text Only*) (Eq. (3)), which excludes image conditioning, and (2) CFG (*Text & Image*), which modifies Eq. (3) to include image conditioning and a unified guidance scale for both modalities. In Fig. 8, methods incorporating reference images improve sample diversity over CFG (Text Only). BCFG achieves higher diversity than CFG (Text & Image) while preserving fidelity, thereby alleviating the trade-off.

## 4.5 Application to Various Pre-Trained Models

We demonstrate the compatibility of our method by applying it to various pre-trained T2I backbones. In Fig. 9, our method achieves the most significant diversity improvement on SDv3.5-Large (Esser et al., 2024), the latest SD version which exhibits the lowest diversity, highlighting its necessity. Furthermore, as newer SOTA T2I models (Esser et al., 2024; DeepFloyd Lab at StabilityAI, 2023; Black Forest Labs, 2024) tend to yield reduced diversity due to increasing model size (Rassin et al., 2024), we demonstrate the effectiveness of our method on the more challenging task of enhancing diversity in models with better original sample diversity (*e.g.*, SDv1.4 and SDv1.1).

## 4.6 Applying BCFG to Various Conditioning Modules

In this section, we examine the proposed Bimodal Classifier-Free Guidance (BCFG) as an independent technique and showcase its applicability across different conditioning modules. Specifically, we apply BCFG to our proposed conditioning module (*e.g.*, Low-Rank Image Adapter) and other image conditioning modules such as *Concatenation + Simple Linear Projection* (LP) (Zhao et al., 2024) and *Decoupled Cross-Attention* (DCA) (Ye et al., 2023). In Tab. 2, we observe that BCFG can enhance both demographic and sample

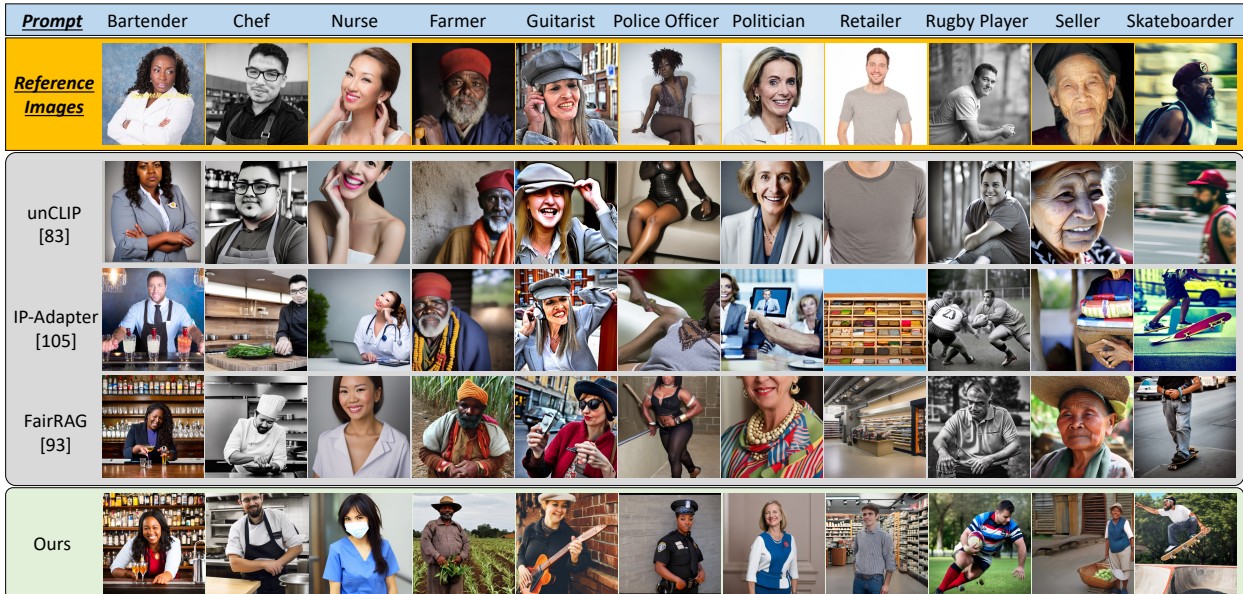

Figure 7: Example outputs from our method and general image-conditioned methods across various occupation categories. Unlike existing methods that often produce images misaligned with the user prompt due to unrelated content in reference images, our method generates prompt-aligned samples, thereby enhancing diversity while maintaining text–image alignment.

*Note.* From left to right, images 1, 3, 5, 6, 7, and 10 use "This person is a female" while the others use "This person is a male" to retrieve the reference image. The parameters used to generate these samples are provided in Appendix 8. The shown samples are illustrative examples selected from a small set of generated outputs. Refer to Tab. 1 for quantitative comparison based on 10,000 images.

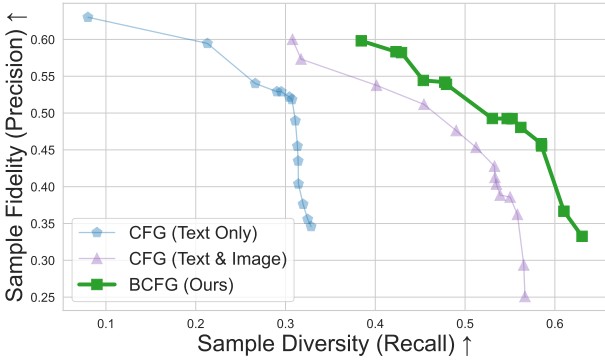

Figure 8: Comparison between guidance methods across different guidance scales for **alleviating the diversity-fidelity trade-off**. Our method achieves higher diversity while preserving fidelity.

diversity across various conditioning modules while maintaining sample fidelity. These results empirically demonstrate that BCFG can be applied independently of the specific conditioning module.

### 4.7 Ablation Study

In Tab. 3, LoRIA enables bimodal conditioning while maintaining prompt alignment, DRS improves demographic diversity, and BCFG enhances sample diversity while preserving fidelity. In Fig. 3, the image guidance scale $\omega_i$ in BCFG, a training-free hyperparameter, can control image modality strength while preserving quality. In contrast, adjusting the image scale in (Ye et al., 2023) or rescaling $\mathbf{e}_i$ during inference

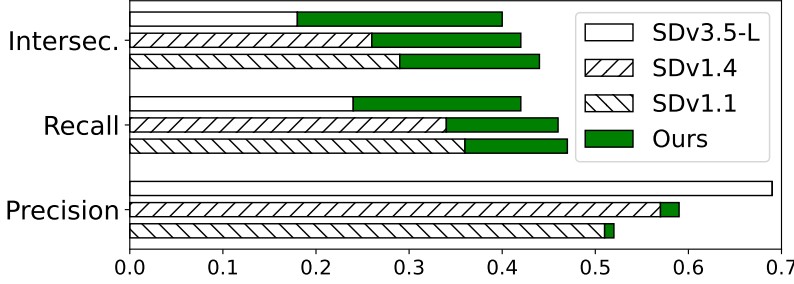

Figure 9: **Application to Various Pre-Trained Models.** Our method achieves the most significant diversity improvement on the latest SD model and even enhances diversity on models with better original sample diversity, all while preserving sample fidelity.

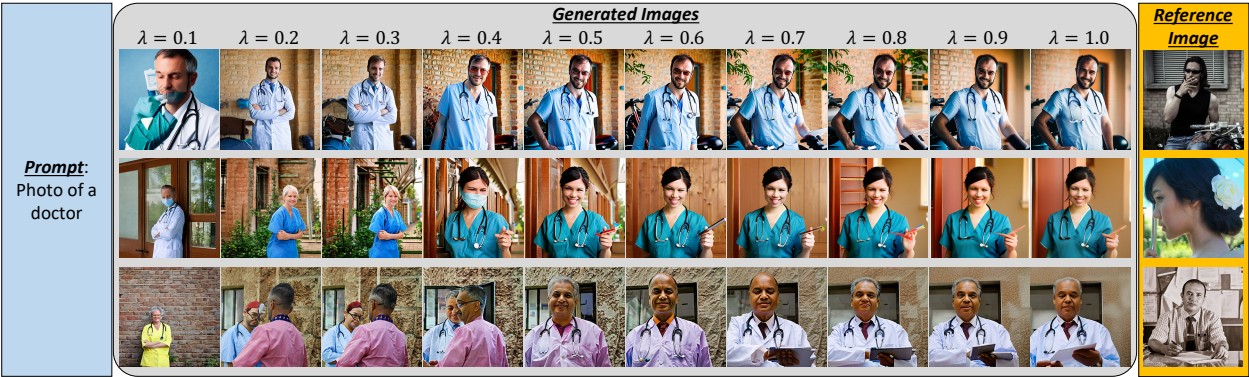

(a) Varying image scale parameter $\lambda^{\text{inference}}$ during inference while keeping image scale parameter $\lambda^{\text{training}}$ fixed at 1.0 during training.

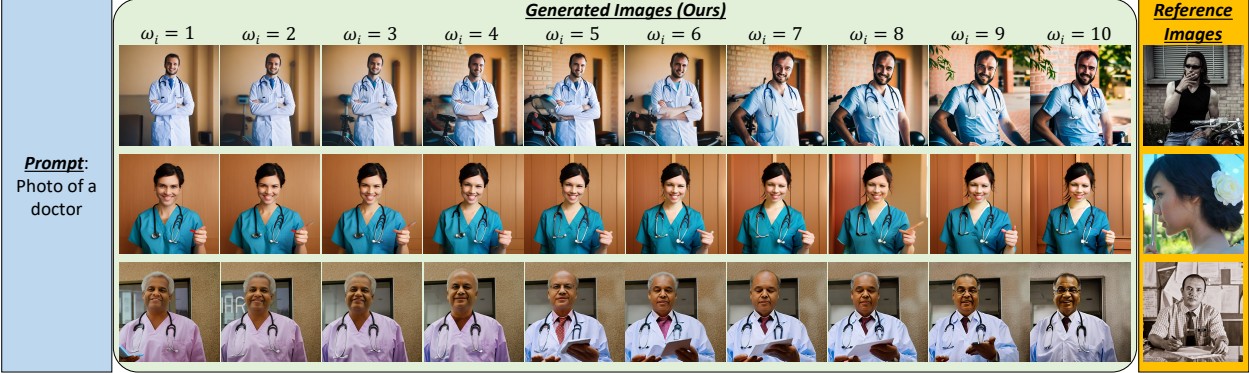

(b) Varying image guidance scale $\omega_i$ in Bimodal Classifier-Free Guidance.

Figure 10: Comparison between control using the image scale parameter $\lambda$ in Decoupled Cross-Attention Ye et al. (2023) and control using the image guidance scale $\omega_i$ in Bimodal Classifier-Free Guidance.

may cause discrepancy between training and inference, leading to overexposure or low resolution (Lin et al., 2024b) (Appendix 3). We also evaluate our method combined with TextAug (*With TextAug*) by conditioning on $(p, d, i)$. This setup slightly increases diversity but compromises fidelity, possibly because reference images already encode demographic information, making the added dummy information by $d$ suboptimal. To compare with TAP, we construct *Details-Augmented Prompt* (DetailsAug) (Esposito et al., 2023; Datta et al., 2023), where we instruct ChatGPT-o1 (OpenAI, 2024) (see instruction in Appendix 5) to generate a detailed text description $dd$ based on the user prompt, including variations in sex, age, skin tone, location,

Table 2: Applying BCFG to Various Conditioning Modules. BCFG can enhance demographic and sample diversity across various conditioning modules while preserving sample fidelity.

| Method | Demographic Diversity ↑ | | | | Sample Diversity | | Sample Fidelity | | Sample Quality | | Alignment |
|---|---|---|---|---|---|---|---|---|---|---|---|
| | Sex | Age | Skin Tone | Intersec. | W1KP ↓ | Recall ↑ | IS ↑ | Precision ↑ | FID ↓ | $F1_{PR}$ ↑ | CLIP ↑ |
| SDv2.1 Rombach et al. (2022) | 0.27 | 0.22 | 0.22 | 0.19 | 0.88 | 0.31 | **22.81** | 0.52 | 27.87 | 0.39 | **23.63** |
| LP (CFG) | 0.75 | 0.51 | 0.40 | 0.44 | 0.60 | 0.43 | 17.13 | 0.42 | 30.70 | 0.42 | 21.33 |
| LP (BCFG) | 0.75 | 0.56 | 0.40 | **0.47** | 0.57 | 0.44 | 20.88 | 0.53 | 25.41 | 0.48 | 23.05 |
| DCA (CFG) | 0.77 | 0.54 | **0.44** | 0.41 | **0.51** | 0.51 | 15.57 | 0.44 | 27.52 | 0.47 | 21.91 |
| DCA (BCFG) | 0.76 | 0.55 | 0.42 | **0.47** | 0.59 | 0.45 | 17.60 | **0.54** | 25.44 | **0.49** | 22.09 |
| LoRIA (CFG) | 0.81 | 0.52 | 0.41 | 0.42 | 0.53 | **0.53** | 16.61 | 0.41 | 26.06 | 0.46 | 22.20 |
| LoRIA (BCFG) | **0.82** | **0.57** | 0.42 | **0.47** | 0.54 | 0.45 | 22.45 | **0.54** | **23.18** | 0.49 | 23.05 |

Table 3: Ablation study of our proposed method.

| Method | Intersec. ↑ | Recall ↑ | Precision ↑ | FID ↓ | CLIP ↑ |
|---|---|---|---|---|---|
| SDv2.1 | 0.19 | 0.31 | 0.52 | 27.87 | **23.63** |
| TextAug | 0.34 | 0.28 | 0.49 | 30.81 | 23.02 |
| DetailsAug | 0.43 | 0.46 | 0.43 | 30.86 | 21.20 |
| *Ablated Variants of Our Method* | | | | | |
| With TextAug | **0.47** | 0.44 | 0.49 | 31.98 | 22.46 |
| W/o LoRIA | **0.47** | 0.44 | 0.53 | 25.41 | 22.17 |
| W/o DRS | 0.18 | 0.44 | 0.53 | 23.58 | 23.39 |
| W/o BCFG | 0.42 | **0.53** | 0.41 | 26.06 | 22.20 |
| Training w/o only $\mathbf{e}_i$ | 0.23 | 0.35 | 0.52 | 26.12 | 23.58 |
| Reference (CelebA) | 0.46 | 0.40 | 0.52 | 25.23 | 23.49 |
| Ours | **0.47** | 0.45 | **0.54** | **23.18** | 23.05 |

and camera settings. Our method $(p, i)$ achieves comparable diversity while preserving FID and CLIP, but DetailsAug $(p, dd)$ compromises text-image alignment, likely because added context in DetailsAug shifts focus from the original user prompt, occasionally leading to unrelated outputs (Hao et al., 2024). Moreover, excluding model training on cases with only visual token $\mathbf{e}_i$ (*Training w/o only* $\mathbf{e}_i$) significantly reduces demographic diversity, dropping Intersectional Diversity from 0.47 to 0.23. This confirms that our training strategy effectively encodes demographic information into $\mathbf{e}_i$, as the *only* $\mathbf{e}_i$ case forces the model to rely solely on images for noise removal, requiring maximal extraction of image information into $\mathbf{e}_i$. Additionally, using lower-quality retrieval databases (*e.g.*, CelebA (Liu et al., 2018)) slightly lowers fidelity and quality but maintains competitive diversity, demonstrating the robustness of our method across various reference databases. In Appendix 4, we present a detailed ablation on the number of retrieval and reference images in DRS, additional BCFG guidance scale results, and an analysis of LoRIA regarding LoRA Rank.

To further study the effects of the retrieval database, we conduct controlled experiments that analyze dataset properties such as size, attribute coverage, gender balance, and intra-group sample diversity. In Tab. 4, the original dataset is denoted as Ours, referring to the full retrieval dataset. Regarding dataset size, we randomly reduce the original dataset to half for comparison. The results show that larger retrieval pools consistently enhance both demographic and sample diversity. Reducing the dataset size also degrades intra-group sample diversity. Thus, having a larger dataset with high sample diversity in the retrieval pool strengthens our

Table 4: Ablation study on the retrieval database.

| Dataset | Sex ↑ | Recall ↑ | Precision ↑ | FID ↓ | CLIP ↑ |
|---|---|---|---|---|---|
| Half Dataset | 0.80 | 0.41 | 0.54 | 23.16 | 23.04 |
| Unbalanced Dataset (Female:Male = 3:1) | 0.62 | 0.38 | 0.53 | 22.42 | 23.05 |
| Balanced Dataset (Female:Male = 1:1) | 0.81 | 0.43 | 0.54 | 22.94 | 23.01 |
| Original Dataset (Ours) | 0.82 | 0.45 | 0.54 | 23.18 | 23.05 |

Table 5: Comparison of our proposed image adapter (Low-Rank Image Adapter) with the existing image adapter (*e.g.*, IP-Adapter) in terms of trainable parameters.

| Method | SD Version | # Trainable Parameters ↓ | FID ↓ | CLIP ↑ |
|---|---|---|---|---|
| IP-Adapter Ye et al. (2023) | v2 | 29.76M | 26.38 | 22.28 |
| | v1 | 21.53M | 28.31 | 22.64 |
| LoRIA (Ours) | v2 | 11.59M | **23.18** | 23.05 |
| | v1 | 8.70M | 24.01 | **23.08** |

method's ability to promote diversity. For attribute coverage and demographic balance, we construct two subsets of the original dataset with female-to-male ratios of 3:1 and 1:1 (balanced). We observe that, due to the limited number of samples in the retrieval dataset, demographic diversity is compromised. Moreover, while the balanced dataset does not further improve performance, it slightly reduces sample diversity because of its smaller size. This observation indicates that balancing the dataset alone has a limited influence on overall diversity. Based on these experiments analyzing the impact of the retrieval database on model performance, we provide several practical guidelines for constructing effective reference databases. First, larger retrieval pools improve both demographic coverage and intra-group sample diversity, which further enhances the diversity of generated samples. Second, ensuring sufficient demographic representation is helpful, as a severe imbalance (*e.g.*, a 3:1 ratio) reduces demographic diversity.

**Study Efficiency.** In this subsection, we analyze the efficiency of the proposed image adapter in terms of the number of trainable parameters, inference steps, and inference latency. By utilizing LoRA, our method achieves bimodal conditioning with minimal parameters. As compared in Tab. 5, our method achieves better image quality and text-image alignment while using fewer trainable parameters. For inference time, the baseline (SDv2.1) and our method take 2.77 seconds and 3.86 seconds, respectively, to generate a single image with 20 denoising steps on a single NVIDIA H100 GPU.

## 5 Conclusion

We propose a lightweight and efficient method to enhance both demographic and sample diversity while preserving fidelity in diffusion models. Extensive empirical results demonstrate its effectiveness in various pre-trained models.

**Limitations and Future Directions.** While our method shows promising results in enhancing demographic diversity, it relies on the assumption that the bias attribute is known. Thus, a potential direction is to extend it to address unknown biases (Li et al., 2022; 2025) by incorporating a bias detection model (*e.g.*, B2T (Kim et al., 2024)) to identify visual biases in T2I models. Moreover, our method leverages IAP to enhance demographic diversity rather than mitigating bias in T2I model itself (Esposito et al., 2023). Another promising direction is to develop T2I models that are less sensitive to biases in the training dataset. Besides, replacing the closed-source database with an open-source one could be beneficial, as it offers more inclusive knowledge to further enhance diversity (Fan et al., 2024).

**Broader Impact Concerns.** In enhancing demographic diversity, we employ templates such as "This person is a [AGE], [SKIN TONE] [SEX]." with corresponding demographic qualifiers. Our aim is not to reduce complex human identities to rigid categories. Instead, we reframe bias mitigation in image generation as a distribution alignment problem, since fairness criteria vary across contexts, and introduce the Adaptable Description Generator (ADG) in Sec. 3.2. ADG is not limited to fixed templates — it can incorporate arbitrary user-defined attributes and fairness criteria, including non-binary representations (as discussed in Appendix 2). In our experiments, we followed existing work Shrestha et al. (2024) for comparability.

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
