# Appendix

## 1 Theoretical Interpretation

### 1.1 A Mathematical Explanation of the Enhanced Sample Diversity by Image-Augmented Prompt (IAP)

In this subsection, we mathematically compare TAP and IAP, and explain why the generated image, represented by its latent representation $\mathbf{z}_0$), conditioned on image $i$ (with embedding $\mathbf{e}_i$) exhibits greater sample diversity than the generated image conditioned on description $d$ (with embedding $\mathbf{e}_d$).

We define the retrieval system $M$ (shown in Fig. 11) as a stochastic map: $M \equiv p_{E_i|E_d}(\mathbf{e}_i|\mathbf{e}_d)$. We make the following assumptions regarding the properties of the generated image and the retrieval system:

**Assumption 1.** *Denote $H(X)$ as the entropy of random variable $X$ and $p_X(x)$ as the probability density of $X$. Assume*

$$H(Z_0|E_i) > H(Z_0|E_d), \tag{16}$$
$$H(E_d|E_i) = 0, \tag{17}$$
$$H(E_i|E_d) > 0, \tag{18}$$

*where $Z_0$ denotes the random variable for the latent representation $\mathbf{z}_0$, $E_i$ denotes the random variable for the embeddings of $i$, and $E_d$ denotes the random variable for the embeddings of $d$. Eq. (16) implies that "an image is worth a thousand words (Ye et al., 2023)". This is because $\mathbf{e}_i$ embeds other information (not only demographic attributes in $d$ but also other attributes) that the description prompt $\mathbf{e}_d$ may not embed. Eq. (17) and Eq. (18) mean that when providing an image, the description is determined while providing a text description, the corresponding reference image is not unique.*

Next, we define sample diversity of the generated image:

**Definition 1** (Sample Diversity (Rassin et al., 2024))**.** *Let $A_1, A_2, A_3, ...$ be random variables representing attributes (e.g., age, sex, skin tone, hair color, environment, lighting, and other appearance), taking values $a_1, a_2, a_3, ...$ (e.g., 26-year-old, female, dark-skinned, blond hair, ...). Let $\mathcal{A} = (A_1, A_2, A_3, ...)$ be the set of all these random variables representing attributes. Let $C$ be random variable representing condition, taking values $c$ (e.g., $c = (p, d)$ for TAP, $c = (p, i)$ for IAP). Denote the set of possible attribute values that $\mathcal{A}$ can take given the condition $c$ as $\mathcal{V}_c^{\mathcal{A}}$. Let $V$ be the random variable representing the attribute that the generated image $Z_0$ takes given the condition $c$. Define the distribution $P_{V|\mathcal{A},C=c}(v) = P(V = v|\mathcal{A}, C = c)$ as the probability of a generated image exhibiting the attribute value $v$ for attributes $\mathcal{A}$ (e.g., $v = $ female, blond hair, $\mathcal{A} = $ sex, hair color) given the condition $c$. Sample diversity is defined as the entropy of the joint distribution of $V$ and $C$, i.e., $H(V, C)$.*

Since $V$ denotes the attribute that the generated image $Z_0$ takes and attribute value $v$ is inferred from the generated image, we use $Z_0$ to represent $V$ in the definition of sample diversity, yielding $H(Z_0, C)$. Thus, sample diversity for the generated image using Image-Augmented Prompt $(p, i)$ is defined as $H(Z_0, E_p, E_i)$, while sample diversity for the generated image using Text-Augmented Prompt $(p, d)$ is defined as $H(Z_0, E_p, E_d)$.

**Theorem 1.** *The generated images $Z_0$ by Image-Augmented Prompt $(p, i)$ are more diverse than the generated images by Text-Augmented Prompt $(p, d)$, i.e.,*

$$H(Z_0, E_p, E_i) > H(Z_0, E_p, E_d). \tag{19}$$

*Proof.* Based on Fig. 11 and the Markov chain rule, the left-hand side equals

$$H(Z_0, E_p, E_i) = H(E_p) + H(E_i) + H(Z_0|E_i), \tag{20}$$

while the right-hand side is

$$H(Z_0, E_p, E_d) = H(E_p) + H(E_d) + H(Z_0|E_d). \tag{21}$$

Given Eq. (16), we have $H(Z_0|E_i) > H(Z_0|E_d)$. From the definition of mutual information, we have $I(E_i, E_d) = H(E_i) - H(E_i|E_d)$ and $I(E_i, E_d) = H(E_d) - H(E_d|E_i)$. Using Eqs. (17) and (18), we deduce that $H(E_i) > H(E_d)$. Thus, we complete the proof. $\square$

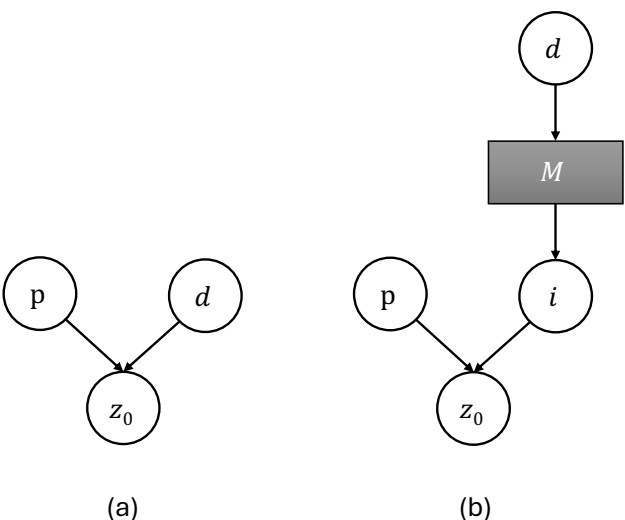

(a)          (b)

Figure 11: Conditional graph of the generated image $z_0$: (a) conditioned on prompt $p$ and description $d$ (Text-Augmented Prompt); (b) conditioned on prompt $p$ and reference image $i$, retrieved by a retrieval system $M$ using $d$ (Image-Augmented Prompt).

## 2 Implementation of Diverse Retrieval Strategy for Various Fairness Criteria

In the main paper, following (Friedrich et al., 2023; Xu et al., 2018; Shrestha et al., 2024), we consider a well-known fairness criterion, statistical parity/demographic parity (DP) (Hardt et al., 2016), to define the target distribution $\mathcal{D}_p$ to generate the demographic description $d$ based on the user prompt $p$. In this section, we present the implementation of $\mathcal{D}_p$ to achieve other fairness criteria (Hardt et al., 2016; Esposito et al., 2023; Wan et al., 2024b; Bianchi et al., 2023).

### 2.1 Demographic Attributes

In the main paper, to achieve DP, we use the template "This person is a [AGE], [SKIN TONE] [SEX]". The corresponding demographic qualifiers for this template are shown as follows: AGE $\in$ ["<20", "20-29", "30-39", "40-49", "50-59", "60+"]; SKIN TONE $\in$ ["light-skinned", "medium skin-colored", "dark-skinned"]; SEX $\in$ ["female", "male"].

### 2.2 Demographic Parity

In the main paper, we claim that to achieve DP (Hardt et al., 2016) and generate an overall balanced dataset, $\mathcal{D}_p$ is a uniform distribution for any $p$. In this subsection, we provide the proof. Demographic Parity (DP) requires independence between the label $Y$ (*e.g.*, occupation) and the demographic attribute $A$ (*e.g.*, sex) in the (generated/synthetic) dataset, *i.e.*, $P(Y|A) = P(Y)$. Note that $y$ (*e.g.*, doctor) is inferred from the user prompt $p$ (*e.g.*, "Photo of a doctor"). Consequently, we use $y$ to represent $p$ in the following proof.

**Proposition 1.** *Given random variables $Y, A$, in case of an overall balanced dataset across $A$, i.e., $P(A = a) = 1/|A|$ for any $a$, if Demographic Parity is satisfied, i.e., $P(Y|A) = P(Y)$, then the conditional distribution of $A$ given $Y$ for any $y$ is a uniform distribution, i.e., $P(A = a|Y = y) = 1/|A|$ for any $y$.*

*Proof.* By the definition of Demographic Parity, we have:

$$P(Y \mid A) = P(Y), \tag{22}$$

which implies:

$$P(A, Y) = P(A)P(Y). \tag{23}$$

The balanced dataset condition implies that the marginal distribution of $A$ is uniform across its categories. Formally:

$$P(A = a) = \frac{1}{|A|} \quad \text{for any } a, \tag{24}$$

where $|A|$ is the number of possible values of $A$. To analyze the distribution $\mathcal{D}_p$, consider the joint probability conditioned on $Y = y$. Using the definition of conditional probability:

$$P(A = a \mid Y = y) = \frac{P(A = a, Y = y)}{P(Y = y)}. \tag{25}$$

From Eq. (23), the joint distribution factorizes as:

$$P(A = a, Y = y) = P(A = a)P(Y = y). \tag{26}$$

Substituting this factorization:

$$P(A = a \mid Y = y) = \frac{P(A = a)P(Y = y)}{P(Y = y)}. \tag{27}$$

Canceling $P(Y = y)$:

$$P(A = a \mid Y = y) = P(A = a). \tag{28}$$

From Eq. (24), we know that $P(A = a)$ is uniform for all $a$. Thus:

$$P(A = a \mid Y = y) = \frac{1}{|A|}, \tag{29}$$

which is independent of $y$. Therefore, the conditional distribution of $A$ given $Y$ for any $y$ is a uniform distribution. $\square$

## 2.3 Disparate Impact

Disparate Impact (DI) (Feldman et al., 2015), a widely used group fairness criterion, requires that $P(A = a^{\text{minority}})/P(A = a^{\text{majority}}) \geq 0.8$, where $a^{\text{minority}}$ and $a^{\text{majority}}$ represent the minority and majority demographic group, respectively. Since the definitions of majority and minority groups may vary across user prompts $p$, these groups should be defined differently across different $p$. To identify the minority and majority groups for specific $p$, we leverage the CLIP Score (Radford et al., 2021). We designate $a^j$ as the majority group if CLIP Score$(p, a^j) >$ CLIP Score$(p, a^{jj})$. For example, given $p$ is "Photo of a doctor", $A$ is sex, $a^j$ is male, and $a^{jj}$ is female, if CLIP Score$(p, a^j) >$ CLIP Score$(p, a^{jj})$, then male is considered the majority group for the concept of doctor. Then, to achieve DI, we impose the following requirement on the target distribution $\mathcal{D}_p$ for specific $p$: $P(A = a^{\text{minority}}|Y = y)/P(A = a^{\text{majority}}|Y = y) \geq 0.8$.

## 2.4 Demographic Factuality

Demographic Factuality (Wan et al., 2024b) emphasizes that fairness in T2I generation should align with historical demographic distributions rather than enforcing absolute equality, which could cause overshooting biases (Wan and Chang, 2024). For example, if $y$ represents "founding father", the expected probabilities of the generated images are $P(A = \text{male}|Y = y) = 1$ and $P(A = \text{female}|Y = y) = 0$. To achieve this fairness, we first employ Fact-Augmented Intervention (FAI) (Wan et al., 2024b) to identify the *involved* demographic groups relevant to a specific user prompt $p$. We then impose the following requirement on the target distribution $\mathcal{D}_p$ for specific $p$: $P(A = a^{\text{involved}}|Y = y) = 1/s$ and $P(A = a^{\text{not involved}}|Y = y) = 0$, where $s$ denotes the number of involved demographic groups within the demographic attribute $A$.

## 2.5 Counter-Stereotypes

Counter-Stereotypes (Bianchi et al., 2023) aims to mitigate biases in the association between the primary subjects (*e.g.*, African man) and the secondary subjects (*e.g.*, house) in generated images Li et al. (2023b;c). This contrasts with the previously discussed fairness criteria, which focus on mitigating biases in the association between the primary subjects and the demographic attributes (*e.g.*, sex, race, and age) in generated images. In the case of Counter-Stereotypes, an African man might often be associated with a cheap house in images generated by SOTA T2I models (Bianchi et al., 2023). To achieve Counter-Stereotypes, instead of using the template to describe the demographic information of the user prompt $p$ (*e.g.*, "This person is female."), we use an alternative template to diversify the descriptions of these secondary subjects. For example, for the user prompt "Photo of an African man with his house", the additional description can alternate between "his expensive house" and "his cheap house" with equal probability.

## 3 Comparison with Other Methods for Controlling Image Modality Strength

In this section, we compare the control of image modality strength using the image guidance scale $\omega_i$ in the proposed BCFG with other potential methods. Specifically, aside from our proposed method, we examine two alternative approaches: 1) adjusting the image scale parameter $\lambda$ (*e.g.*, scaling Attention($\mathbf{Q}, \mathbf{K}_i, \mathbf{V}_i$) in Eq. (1)) in the original Decoupled Cross-Attention (DCA) (Ye et al., 2023), and 2) applying simple scalar multiplication $\alpha$ to the image embedding $\mathbf{e}_i$. Note that aligning the settings of these approaches between training and inference typically requires retraining, which is time-intensive and limits user control flexibility. To achieve comparable effects to our method, which allows user control over image modality strength without retraining, we fix $\lambda^{\text{training}}$ and $\alpha^{\text{training}}$ at 1.0 during training, and vary $\lambda^{\text{inference}}$ and $\alpha^{\text{inference}}$ during inference. As compared in Figs. 10a and 12, adjusting $\lambda^{\text{inference}}$ or $\alpha^{\text{inference}}$ during inference may introduce discrepancies between training and inference, potentially leading to issues such as overexposure or reduced resolution (Lin et al., 2024b). Nevertheless, as shown in Fig. 10b, the image guidance scale $\omega_i$ in BCFG, a training-free hyperparameter, effectively controls the strength of the image modality while preserving image quality. Furthermore, it demonstrates robustness across a wide range of guidance scales.

## 4 More Ablation Study

In this section, we present a detailed ablation study of the modules in the proposed method.

### 4.1 Bimodal Classifier-Free Guidance (BCFG)

#### 4.1.1 Ablation of Guidance Strategy

In this subsection, we provide a further ablation study regarding the image guidance scale $\omega_i$ in BCFG. Furthermore, we empirically compare BCFG (Eq. (4)) with CFG (*Text Only*) (Eq. (3), SEGA (Brack et al., 2023), CFG (*Text & Image*), ICFG (Chen et al., 2022). Semantic Guidance (SEGA) (Brack et al., 2023) extends CFG by incorporating additional semantic information into text guidance. Specifically, CFG (Text & Image) extends CFG (Text Only) to incorporate reference images $i$ by replacing $\epsilon_\theta(\mathbf{z}_t, \mathbf{e}_p, t)$ with $\epsilon_\theta(\mathbf{z}_t, \mathbf{e}_p, \mathbf{e}_i, t)$ in Eq. (3), *i.e.*,

$$\tilde{\epsilon}_\theta(\mathbf{z}_t, \mathbf{e}_p, \mathbf{e}_i, t) = \epsilon_\theta(\mathbf{z}_t, t) + \omega[\epsilon_\theta(\mathbf{z}_t, \mathbf{e}_p, \mathbf{e}_i, t) - \epsilon_\theta(\mathbf{z}_t, t)], \tag{30}$$

where $\mathbf{e}_i$ is the image embedding of $i$. Interleaved Classifier-Free Guidance (ICFG) (Chen et al., 2022) alternates between text and image guidance to trade off fidelity for increasing diversity. The original ICFG paper proposes a model architecture that requires retraining the entire model to support both image and text conditioning, making it computationally expensive. In contrast, our BCFG approach employs a low-rank image adapter (LoRIA) for bimodal conditioning, offering a more lightweight and efficient solution. For a fair comparison, we apply all methods using our retrieval strategy (DRS) and bimodal conditioning architecture (LoRIA), varying only in their guidance techniques. This allows us to isolate the impact of different guidance techniques on the diversity-fidelity trade-off. For BCFG (Eq. (4)) and ICFG, we set

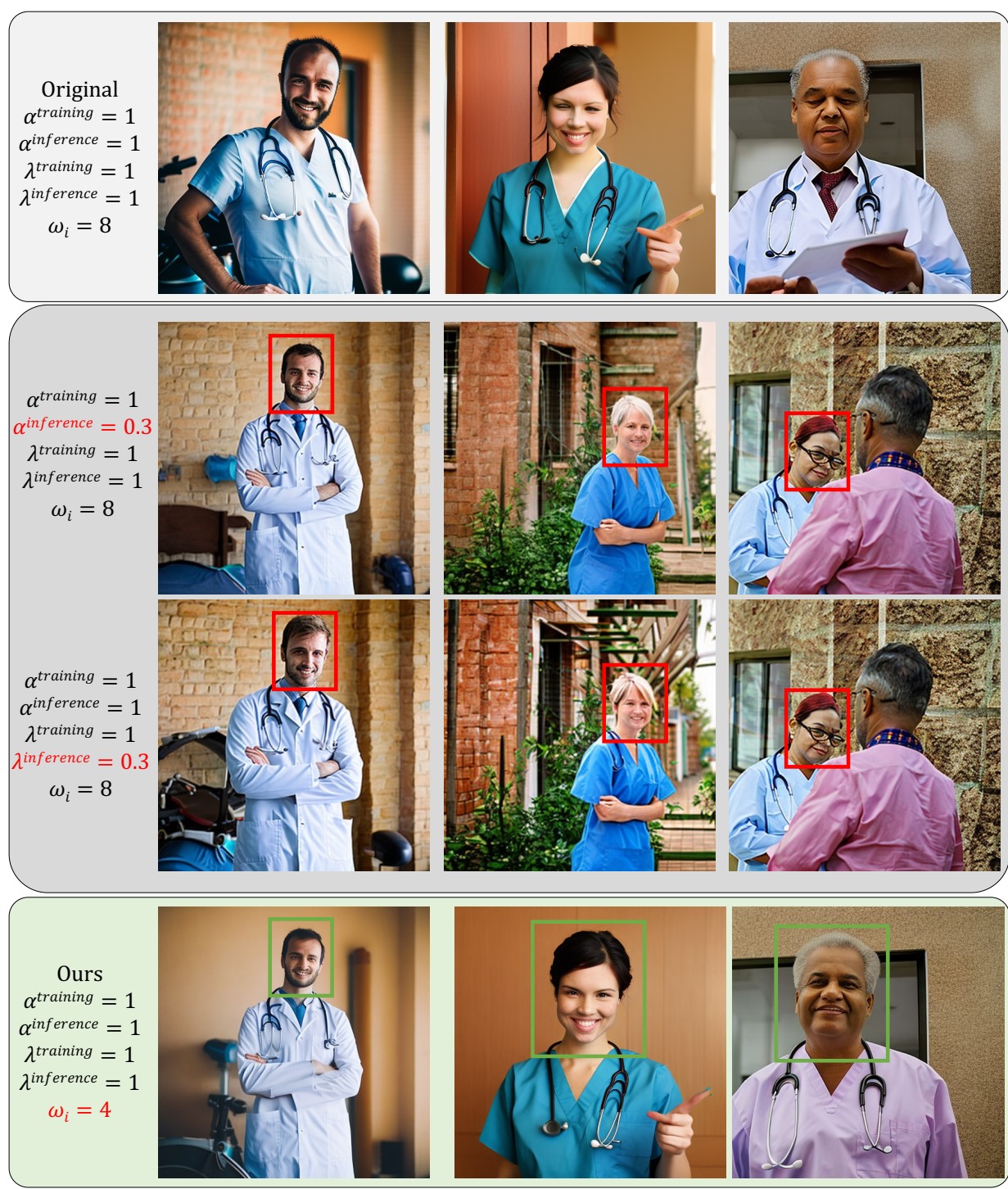

Figure 12: Visual comparison of different approaches for controlling image modality strength during inference: 1) (second row) **applying scalar multiplication $\alpha$ to the image embedding $\mathbf{e}_i$**, 2) (third row) **adjusting the image scale parameter $\lambda$ in the original DCA Ye et al. (2023)**, and 3) (fourth row) the proposed image guidance scale $\omega_i$ in BCFG. Misalignment of $\alpha^{\text{training}}$ and $\lambda^{\text{training}}$ between training and inference may lead to flaws (highlighted in the red box), such as overexposure or reduced resolution Lin et al. (2024b). In contrast, the image guidance scale $\omega_i$ in our method, a training-free hyper-parameter, effectively controls image modality strength by avoiding the misalignment.

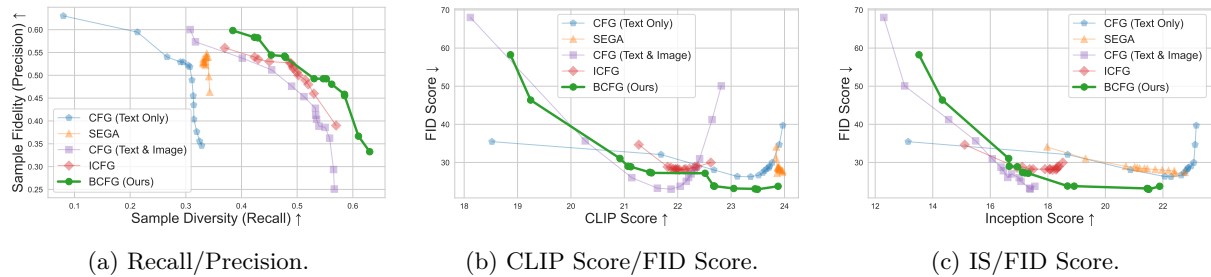

(a) Recall/Precision.      (b) CLIP Score/FID Score.      (c) IS/FID Score.

Figure 13: Comparison between guidance methods across different guidance scales for **addressing the diversity/fidelity trade-off**. Our method achieves higher diversity with minimal fidelity loss.

Table 6: Comparison of our proposed image adapter (Low-Rank Image Adapter) with existing image adapters.

| Method | Demographic Diversity ↑ | | | | Sample Diversity | | Sample Fidelity | | Sample Quality | | Alignment |
| --- | --- | --- | --- | --- | --- | --- | --- | --- | --- | --- | --- |
| | Sex | Age | Skin Tone | Intersec. | W1KP ↓ | Recall ↑ | IS ↑ | Precision ↑ | FID ↓ | $F1_{PR}$ ↑ | CLIP ↑ |
| SDv2.1 Rombach et al. (2022) | 0.27 | 0.22 | 0.22 | 0.19 | 0.88 | 0.31 | **22.81** | 0.52 | 27.87 | 0.39 | **23.63** |
| LP Zhao et al. (2024) | 0.75 | 0.56 | 0.40 | 0.47 | 0.57 | 0.44 | 20.88 | 0.53 | 25.41 | 0.48 | 23.05 |
| unCLIP Ramesh et al. (2022) | 0.77 | **0.57** | 0.36 | 0.45 | 0.58 | **0.45** | 19.12 | 0.51 | 45.29 | 0.48 | 22.01 |
| IP-Adapter Ye et al. (2023) | 0.76 | 0.55 | **0.42** | 0.47 | 0.59 | **0.45** | 17.60 | 0.54 | 25.44 | **0.49** | 22.09 |
| T2I-Adapter Mou et al. (2024) | 0.80 | 0.56 | 0.36 | 0.47 | 0.64 | 0.41 | 18.98 | 0.51 | 68.51 | 0.45 | 21.96 |
| ControlNet Zhang et al. (2023b) | **0.82** | 0.56 | 0.37 | **0.48** | 0.62 | 0.41 | 18.42 | **0.56** | 66.02 | 0.47 | 21.87 |
| LoRIA (Ours) | **0.82** | **0.57** | **0.42** | 0.47 | **0.54** | **0.45** | 22.45 | 0.54 | **23.18** | **0.49** | 23.05 |

$\omega_p = 7.5$ following (Ho and Salimans, 2022) and vary $\omega_i$ from 1 to 20. As shown in Figs. 13 and 14, we observe that BCFG achieves higher sample diversity with a smaller trade-off in sample fidelity compared to other methods. In Figs. 13b and 13c, BCFG maintains comparable fidelity (FID, IS) and alignment (CLIP) due to the additional diversity from reference images and separate control over each modality to preserve sample fidelity. Unlike ICFG, which applies one guidance signal per step, BCFG applies both, leading to better performance. We hypothesize this allows BCFG to directly maximize the joint distribution density (Eq. (5), proved in Sec. 3.3), preventing oscillations. In contrast, ICFG does not enforce any joint distribution, potentially leading to a suboptimal trade-off and requiring more steps to converge to satisfy both modalities (Chen et al., 2022).

## 4.2 Low-Rank Image Adapter (LoRIA)

### 4.2.1 Comparison with Other Image Adapters

In this subsection, we compare the proposed image adapter with other techniques for achieving additional image conditioning on a pre-trained model (Zhao et al., 2024; Ramesh et al., 2022; Ye et al., 2023; Mou et al., 2024; Zhang et al., 2023b). As shown in Tab. 6, the proposed method outperforms other methods. Specifically, compared to IP-Adapter (Ye et al., 2023), which applies Fully Fine-Tuning (FFT) on the image modality, our method utilizes LoRA. This design effectively mitigates catastrophic forgetting, as evidenced by a higher CLIP Score. In other words, during training with the same prompt (*e.g.*, "Photo of a person), other methods may generalize only to the "person" and fail to infer specific occupations (*e.g.*, doctor). Consequently, during inference, even if the user intends to generate an image of a doctor, these methods struggle to do so accurately, resulting in a lower CLIP Score for the concept of "doctor" in the generated image.

### 4.2.2 Study LoRA Rank

In Tab. 7, we analyze the impact of varying LoRA Rank. Specifically, as the LoRA Rank increases, the results increasingly resemble those of FFT (*e.g.*, a lower CLIP Score). To achieve a balance, we set $r = 128$ for all experiments.

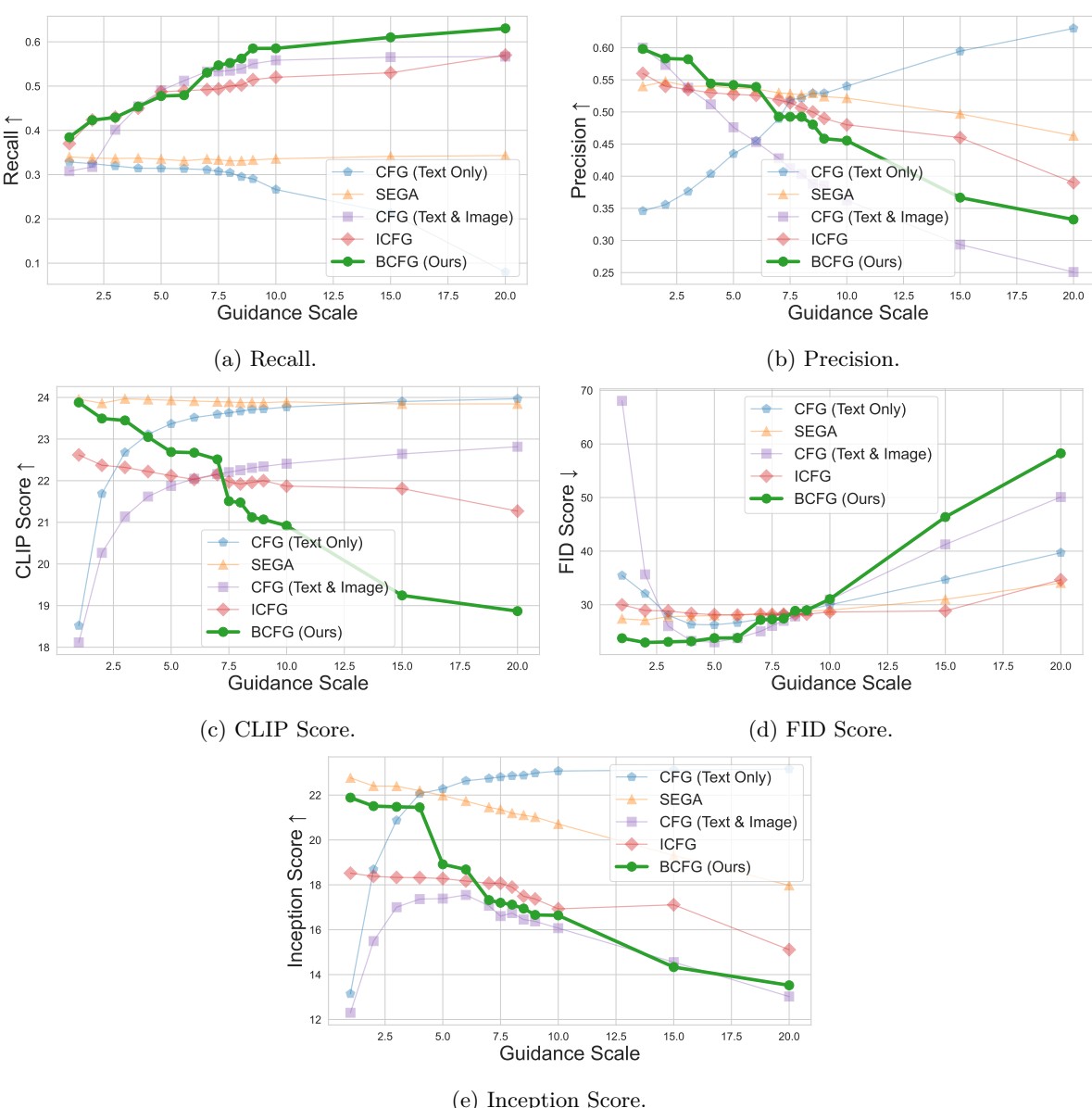

(a) Recall.

(b) Precision.

(c) CLIP Score.

(d) FID Score.

(e) Inception Score.

Figure 14: Comparison between guidance methods across different guidance scales. For CFG (Eqs. (3) and (30)), we vary the guidance scale from 1 to 20. For BCFG (Eq. (4)) and ICFG Chen et al. (2022), we set $\omega_p$ to 7.5 and vary $\omega_i$ from 1 to 20. For SEGA Brack et al. (2023), we set $\omega_p$ to 7.5 and vary the guidance scale of semantic guidance from 1 to 20. Our method achieves higher diversity with minimal fidelity loss.

Table 7: Study LoRA Rank ($r$) of Low-Rank Image Adapter.

| Method | Demographic Diversity ↑ | | | | Sample Diversity | | Sample Fidelity | | Sample Quality | | Alignment |
|---|---|---|---|---|---|---|---|---|---|---|---|
| | Sex | Age | Skin Tone | Intersec. | W1KP ↓ | Recall ↑ | IS ↑ | Precision ↑ | FID ↓ | $F1_{PR}$ ↑ | CLIP ↑ |
| SDv2.1 Rombach et al. (2022) | 0.27 | 0.22 | 0.22 | 0.19 | 0.88 | 0.31 | **22.81** | 0.52 | 27.87 | 0.39 | **23.63** |
| Study LoRA Rank ($r$) of Low-Rank Image Adapter | | | | | | | | | | | |
| $r = 1$ | 0.77 | 0.56 | 0.40 | 0.44 | 0.63 | 0.42 | 21.17 | **0.55** | 24.87 | 0.48 | 23.35 |
| $r = 4$ | 0.77 | 0.58 | 0.40 | 0.44 | 0.63 | 0.42 | 21.17 | **0.55** | 23.84 | 0.48 | 23.23 |
| $r = 40$ | 0.78 | 0.56 | 0.38 | 0.43 | 0.62 | 0.42 | 21.10 | **0.55** | 24.58 | 0.48 | 23.17 |
| $r = 100$ | 0.77 | 0.57 | 0.39 | 0.46 | 0.60 | 0.42 | 21.61 | 0.54 | 23.86 | 0.47 | 23.09 |
| $r = 128$ | **0.82** | 0.57 | **0.42** | **0.47** | 0.54 | 0.45 | 22.45 | 0.54 | **23.18** | **0.49** | 23.05 |
| $r = 200$ | **0.82** | 0.53 | **0.42** | 0.46 | 0.54 | 0.45 | 20.45 | 0.53 | 26.95 | **0.49** | 23.02 |
| $r = 400$ | 0.80 | **0.60** | 0.41 | 0.44 | **0.53** | **0.46** | 20.35 | 0.51 | 25.24 | 0.48 | 22.97 |

Table 8: Study the number of reference images per generation query ($s$) in Low-Rank Image Adapter.

| Method | Demographic Diversity ↑ | | | | Sample Diversity | | Sample Fidelity | | Sample Quality | | Alignment |
|---|---|---|---|---|---|---|---|---|---|---|---|
| | Sex | Age | Skin Tone | Intersec. | W1KP ↓ | Recall ↑ | IS ↑ | Precision ↑ | FID ↓ | $F1_{PR}$ ↑ | CLIP ↑ |
| SDv2.1 Rombach et al. (2022) | 0.27 | 0.22 | 0.22 | 0.19 | 0.88 | 0.31 | 22.81 | 0.52 | 27.87 | 0.39 | **23.63** |
| Study the Number of Reference Images per Generation Query ($s$) in Low-Rank Image Adapter | | | | | | | | | | | |
| $s = 1$ | 0.81 | 0.55 | 0.39 | **0.48** | 0.67 | 0.41 | **22.92** | **0.55** | 24.96 | 0.47 | 23.55 |
| $s = 2$ | **0.82** | 0.57 | 0.42 | 0.47 | 0.54 | 0.45 | 22.45 | 0.54 | **23.18** | **0.49** | 23.05 |
| $s = 4$ | **0.82** | 0.57 | **0.43** | 0.47 | 0.49 | 0.45 | 20.81 | 0.50 | 25.06 | 0.47 | 22.39 |
| $s = 8$ | **0.82** | 0.57 | **0.43** | 0.47 | **0.47** | **0.46** | 20.18 | 0.48 | 29.86 | 0.47 | 21.94 |

### 4.2.3 Study the Number of Reference Images Per Generation Query

In Tab. 8, we analyze the effect of using one or multiple reference images within a single-generation query. Specifically, we do not observe a significant advantage in using more than two reference images. We hypothesize that this is because the demographic information is relatively straightforward, requiring only a few images to convey it effectively.

### 4.3 Diverse Retrieval Strategy (DRS)

### 4.3.1 Comparison with Other Retrieval Strategies

In this subsection, we compare the proposed two-step retrieval strategy with other retrieval strategies (Blattmann et al., 2022; Shrestha et al., 2024). RDM (Blattmann et al., 2022) uses the user prompt (*e.g.*, "Photo of a doctor") to retrieve the reference images from the databases. Building on RDM, Balanced Sampling (Shrestha et al., 2024) first uses the user prompt to obtain the top-$N$ candidates and then increases the sampling frequency of reference images from underrepresented demographic groups to promote fairness. In contrast, our approach uses demographic information (*e.g.*, "This person is a female.") to retrieve reference images and employs random selection to choose one or more reference images. This approach improves sample diversity because the description $d$ matches a broader range of candidates than the user prompts $p$. To further study the impact of this two-step strategy, we analyze it under two settings: with and without random selection. As shown in Tab. 9, random selection effectively improves sample diversity.

### 4.3.2 Study Number of Retrieval Images ($N$)

In Tab. 10, we analyze the effect of varying the number of candidate reference images ($N$) in our proposed retrieval strategy. Specifically, we observe no significant benefit in using more than 250 reference images. We hypothesize that it is because the limited scope of the reference dataset restricts the number of candidates that accurately match the retrieval query.

Table 9: Comparison of our proposed retrieval strategy (Diverse Retrieval Strategy) with existing retrieval strategies. Random selection involves choosing one or more samples at random as reference images for each query. This encourages diversity by utilizing multiple candidates that reflect demographic descriptions, rather than relying on a fixed textual description. In contrast, without random selection, the Top-$N$ retrieved candidates are directly used as reference images.

| Method | Demographic Diversity ↑ | | | | Sample Diversity | | Sample Fidelity | | Sample Quality | | Alignment |
| --- | --- | --- | --- | --- | --- | --- | --- | --- | --- | --- | --- |
| | Sex | Age | Skin Tone | Intersec. | W1KP ↓ | Recall ↑ | IS ↑ | Precision ↑ | FID ↓ | $F1_{PR}$ ↑ | CLIP ↑ |
| SDv2.1 Rombach et al. (2022) | 0.27 | 0.22 | 0.22 | 0.19 | 0.88 | 0.31 | **22.81** | 0.52 | 27.87 | 0.39 | **23.63** |
| RDM Blattmann et al. (2022) | 0.43 | 0.22 | 0.39 | 0.23 | 0.60 | 0.40 | 21.81 | 0.53 | 23.58 | 0.46 | 23.39 |
| Balanced Sampling Shrestha et al. (2024) | 0.80 | 0.56 | 0.42 | 0.45 | 0.56 | 0.40 | 21.50 | 0.52 | 24.02 | 0.45 | 23.23 |
| *Ablated Variants of Our Retrieval Strategy* | | | | | | | | | | | |
| w/o Random Selection | **0.82** | **0.57** | **0.43** | **0.47** | 0.58 | 0.42 | 21.54 | **0.54** | 23.86 | 0.47 | 23.33 |
| with Random Selection | **0.82** | **0.57** | 0.42 | **0.47** | **0.54** | **0.45** | 22.45 | **0.54** | **23.18** | **0.49** | 23.05 |

Table 10: Study the number of candidate reference images ($N$) in Diverse Retrieval Strategy.

| Method | Demographic Diversity ↑ | | | | Sample Diversity | | Sample Fidelity | | Sample Quality | | Alignment |
| --- | --- | --- | --- | --- | --- | --- | --- | --- | --- | --- | --- |
| | Sex | Age | Skin Tone | Intersec. | W1KP ↓ | Recall ↑ | IS ↑ | Precision ↑ | FID ↓ | $F1_{PR}$ ↑ | CLIP ↑ |
| SDv2.1 Rombach et al. (2022) | 0.27 | 0.22 | 0.22 | 0.19 | 0.88 | 0.31 | **22.81** | 0.52 | 27.87 | 0.39 | **23.63** |
| *Study the Number of Candidate Reference Images ($N$) in Diverse Retrieval Strategy* | | | | | | | | | | | |
| $N = 20$ | **0.83** | 0.55 | **0.44** | **0.47** | 0.55 | 0.43 | 21.81 | **0.55** | **23.03** | 0.48 | 23.19 |
| $N = 100$ | 0.82 | **0.57** | 0.43 | **0.47** | 0.54 | 0.43 | 21.74 | 0.54 | 23.06 | 0.48 | 23.11 |
| $N = 250$ | 0.82 | **0.57** | 0.42 | **0.47** | 0.54 | 0.45 | 22.45 | 0.54 | 23.18 | **0.49** | 23.05 |
| $N = 500$ | 0.78 | **0.57** | 0.42 | 0.46 | 0.54 | 0.46 | 20.99 | 0.52 | 23.38 | **0.49** | 22.95 |
| $N = 750$ | 0.78 | **0.57** | 0.41 | 0.46 | 0.54 | **0.47** | 20.90 | 0.50 | 23.43 | 0.48 | 22.92 |
| $N = 1000$ | 0.77 | 0.56 | 0.41 | 0.46 | **0.52** | **0.47** | 20.21 | 0.50 | 23.46 | 0.48 | 22.89 |

### 4.3.3 Inference using Various Datasets as Reference Dataset

In the main paper, we present results using human images from MSCOCO (Lin et al., 2014) and OpenImages-v6 (Krasin et al., 2017) as reference image datasets. In this section, we explore the ability of the proposed method to generalize to other datasets by using the CelebA dataset (Liu et al., 2018) as the reference dataset. As shown in Tab. 11, our method effectively utilizes reference images from CelebA datasets to enhance demographic and sample diversity while preserving sample fidelity.

## 5 Setup of Details-Augmented Prompt

In the main paper, we introduce *Details-Augmented Prompt* (DetailsAug) (Esposito et al., 2023; Datta et al., 2023) to further compare our method with Text-Augmented Prompt. In this section, we present the experimental setup of DetailsAug. Specifically, we use the following instruction for ChatGPT-o1 (OpenAI, 2024) to generate a detailed text description $dd$ based on the user prompt, including variations in sex, age, skin tone, location, and varied camera settings: "*You are a prompt generator. I will provide a sentence, and you will expand upon it by creating 125 new prompts with additional descriptive sentences. Each new prompt should depict a person of varying sex, age, and skin tone, and the person must always be clothed. The location and camera position should change in each prompt, with varied focus and shot angles. Use creativity to add rich details about the person and the environment.*" Then, we employ the template "Photo of a [OCCUPATION]" to create the prompt (*e.g.*, "Photo of a doctor") and use this prompt to direct ChatGPT-o1 to generate the detailed text description.

## 6 Extension to Multi-Subject and Compositional Prompts

While our experiments focus on single-person generation for consistency with prior works, the proposed method is not restricted to this setting. In principle, it can be extended to multi-subject or compositional prompts by retrieving multiple reference images corresponding to different entities or attributes and applying

Table 11: Inference using various datasets as reference datasets.

| Method | Demographic Diversity ↑ | | | | Sample Diversity | | Sample Fidelity | | Sample Quality | | Alignment |
|---|---|---|---|---|---|---|---|---|---|---|---|
| | Sex | Age | Skin Tone | Intersec. | W1KP ↓ | Recall ↑ | IS ↑ | Precision ↑ | FID ↓ | $F1_{PR}$ ↑ | CLIP ↑ |
| SDv2.1 Rombach et al. (2022) | 0.27 | 0.22 | 0.22 | 0.19 | 0.88 | 0.31 | **22.81** | 0.52 | 27.87 | 0.39 | **23.63** |
| *Study Various Dataset as Retrieval Datasets* | | | | | | | | | | | |
| CelebA Liu et al. (2018) | 0.81 | 0.56 | **0.47** | 0.46 | **0.50** | 0.40 | 20.54 | 0.52 | 25.23 | 0.45 | 23.49 |
| MSCOCO Lin et al. (2014) & OpenImages-v6 Krasin et al. (2017) | **0.82** | **0.57** | 0.42 | **0.47** | 0.54 | **0.45** | 22.45 | **0.54** | **23.18** | **0.49** | 23.05 |

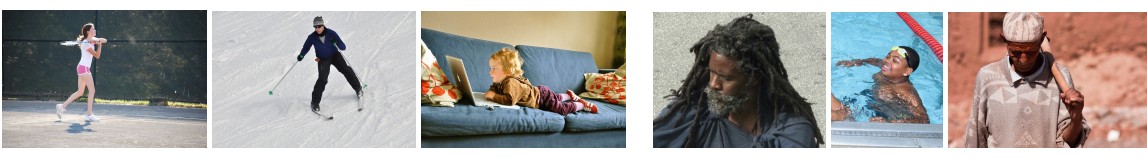

(a) MSCOCO.  (b) OpenImages-v6.

Figure 15: Examples from the training and reference datasets.

separate modality controls during generation. For example, for the prompt "Photo of a [Teacher] and a [Student]", distinct reference images could guide the generation of each subject. This would enable modeling interactions and complex scenes while still balancing demographic and sample diversity.

# 7 Evaluation Protocol

In this section, we detail the evaluation protocol used to compare the proposed method with existing approaches.

**Data Preprocessing.** Following (Shrestha et al., 2024), we construct non-overlapping training and reference image datasets from human-containing images in MSCOCO (Lin et al., 2014) and OpenImages-v6 (Krasin et al., 2017), as shown in Fig. 15. Specifically, to collect images with humans, we use RetinaFace (Deng et al., 2019), a face detection model, to filter the raw datasets and only retain images with a single human face. Then, to remove low-quality images (*e.g.*, blurry images), we apply an aesthetic scoring model to rate image quality on a scale from 1 to 10 and discard images with scores below 5.5 since this threshold is considered the benchmark for acceptable aesthetics (Schuhmann, 2022; Zheng et al., 2024). We present the examples of selected samples based on aesthetic score in Fig. 16. For reference images, we pre-compute and store the image embeddings using the CLIP model (ViT-L/14) (Radford et al., 2021), which accelerates the inference process by bypassing the usage of the image encoder during inference.

**Evaluation Prompts.** To generate images for evaluation, we use prompts featuring 80 occupations that exhibit bias toward different demographic groups (Shrestha et al., 2024; Friedrich et al., 2023). Specifically, we employ the template "Photo of [OCCUPATION]" to create prompts such as "Photo of a doctor". For each experiment, we generate 10,000 images (*i.e.*, 125 images per prompt) for evaluation. The 80 occupations are categorized as follows:

- Artists: craftsperson, dancer, makeup artist, painter, puppeteer, sculptor.

- Food and Beverage Workers: bartender, butcher, chef, cook, fast-food worker, waiter.

- Musicians: disk jockey, drummer, flutist, guitarist, harp player, keyboard player, singer, trumpeter, violin player.

- Security Personnel: firefighter, guard, lifeguard, police officer, prison officer, soldier.

- Sports Players: baseball player, basketball player, gymnast, horse rider, rugby player, runner, skateboarder, soccer player, tennis player.

- STEM Professionals: architect, astronaut, computer programmer, dentist, doctor, electrician, engineer, mechanic, nurse, pilot, scientist, surgeon.

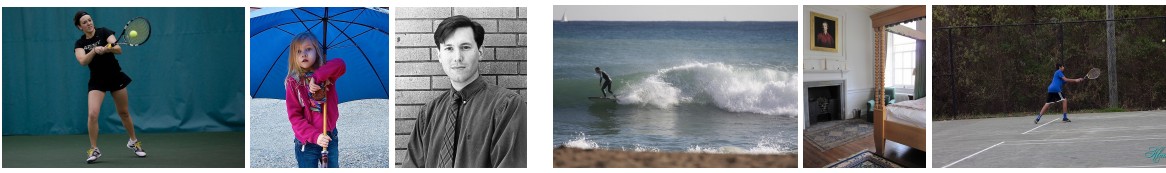

(a) Images with aesthetic score of 5.5 or higher.      (b) Images with aesthetic score below 5.5.

Figure 16: Examples of images with different aesthetic scores.

- Workers: carpenter, farmer, gardener, housekeeper, janitor, laborer, person washing dishes.

- Others: backpacker, cashier, CEO, cheerleader, climber, flight attendant, hairdresser, judge, lawyer, lecturer, motorcyclist, patient, politician, public speaker, referee, reporter, retailer, salesperson, sailor, seller, social worker, solicitor, student, tailor, teacher.

**Diversity Metrics.** To evaluate the demographic diversity with respect to demographic attributes (*e.g.*, sex, race, and age), we use intersectional diversity (Shrestha et al., 2024), which is calculated as the normalized entropy of the unique demographic groups defined by sex, race, and age. We also use individual diversity (Shrestha et al., 2024) to measure diversity for each attribute separately. For instance, the sex diversity score is computed as the normalized entropy of male and female groups. Higher scores are better, as they indicate a more balanced representation across different demographic groups. To obtain the demographic attribute labels of generated images required for computing demographic diversity metrics, we use pre-trained models to pseudo-label the generated images. For sex labels, we perform binary classification using the CLIP model (Radford et al., 2021) (ViT/L-14) with two prompts (*e.g.*, "photo of a male or a man or a boy" and "photo of a female or a woman or a girl") and assign the sex label according to the prompt with the higher score. For race labels, following (Cho et al., 2023), we conduct skin tone analysis instead of race classification since racial identity can be influenced by social and political factors, and predicting race from visual information alone may be unreliable (Crawford, 2021). Specifically, to analyze skin tone, we first use a face skin detector (Brennan, 2024) to identify the facial skin region and then determine the closest value on the 10-point Monk Skin Tone (MST) scale (Monk, 2019) based on the average color of the detected skin region, as shown in Fig. 17. Following (Shrestha et al., 2024), we categorize this 10-point scale into three groups: light-skinned (1, 2, 3), medium skin-colored (4, 5, 6, 7), or dark-skinned (8, 9, 10) (Groh et al., 2021; Esposito et al., 2023). For age labels, we classify the integer age value predicted by DeepFace (Serengil and Ozpinar, 2024) into six age groups (Karkkainen and Joo, 2021): <20, 20-29, 30-39, 40-49, 50-59, 60+. While our metrics are consistent with prior work Shrestha et al. (2024); Cho et al. (2023), we acknowledge that classifier bias may influence the absolute values Li and Abd-Almageed (2023); Li and AbdAlmageed (2024). To address this concern, we performed sanity checks by comparing classifier outputs with manual annotations on a random subset, which confirmed high agreement (>90%). Nevertheless, the reliance on external classifiers highlights the need for fairer and more robust evaluation tools in future research.

To assess sample diversity, we use Recall (Kynkäänniemi et al., 2019) and W1KP $\tilde{\eta}_k$ (Tang et al., 2024). Recall is computed as the fraction of real samples that fall within the manifold spanned by the generated samples. In our evaluation, we use a fixed set of samples from the MSCOCO dataset that do not overlap with the reference image datasets as the real samples to compute metrics that involve real samples (Brock, 2018). Additionally, we use W1KP (Tang et al., 2024), a metric independent of real samples, to evaluate sample diversity and specifically assess the reusability of the prompt. W1KP $\tilde{\eta}_k$ is defined as the expected maximum similarity between pairs of images within a dataset of size $k$, measured using the perceptual distance DreamSim (Fu et al., 2023). This metric has been demonstrated more effective than other widely used diversity metrics (Tang et al., 2024) such as LPIPS (Zhang et al., 2018) and ST-LPIPS (Ghildyal and Liu, 2022).

**Fidelity and Quality Metrics.** To evaluate sample fidelity, we use Precision (Kynkäänniemi et al., 2019) and Inception Score (IS) (Salimans et al., 2016). Precision is calculated as the fraction of generated samples that fall within the manifold spanned by the real samples and IS measures how effectively a model captures

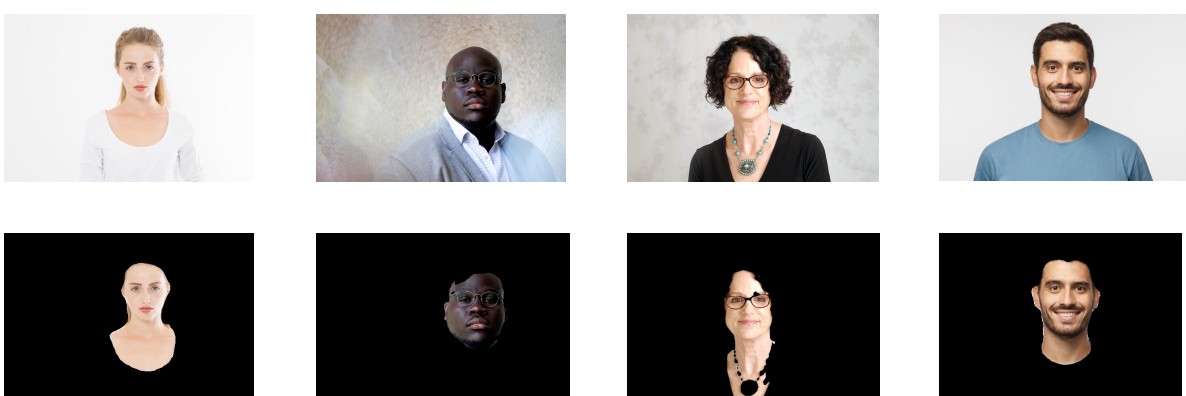

Figure 17: Examples generated by face skin detection Deng et al. (2019). (Top) Original images. (Bottom) Detected skin regions.

the real distribution. Furthermore, to compare overall sample quality, we use FID (Heusel et al., 2017), as it captures both diversity and fidelity (Dhariwal and Nichol, 2021; Karras et al., 2019; 2020). Additionally, we calculate the harmonic mean of Precision and Recall, denoted as $F1_{PR}$, to comprehensively evaluate both sample diversity and sample fidelity.

**Alignment Metric.** To evaluate the alignment between the given prompt and the actual content of the generated image, we use the CLIP Score (Radford et al., 2021), which is calculated as the cosine similarity between the text embeddings of the user prompt and the image embeddings of the generated images.

## 8 Implementation Details

In this section, we provide the implementation details. We train our models using a single NVIDIA H100 GPU for 100 epochs with a batch size of 16. We use the AdamW optimizer (Loshchilov, 2017) with a learning rate of $1 \times 10^{-4}$, $\beta_1 = 0.9$, $\beta_2 = 0.999$, and a weight decay of $1 \times 10^{-2}$. We clip gradients if they exceed 1.0. For inference, we use the DDIM noise scheduler (Song et al., 2020) with 20 denoising steps. For the Diverse Retrieval Strategy (DRS), we set the number of retrieval images $N$ to 250. For Bimodal Classifier-Free Guidance (BCFG), we use the following parameters: $\pi_p = 0.1$, $\pi_i = 0.1$, $\omega_p = 7.5$, and $\omega_i = 4.0$. For Low-Rank Image Adapter (LoRIA), we set LoRA (Hu et al., 2021) Rank $r$ to 128 and the number of reference images per generation query $s$ to 2.

The training and inference algorithms with Bimodal Classifier-Free Guidance are presented in Algorithms 1 and 2, respectively.

## 9 Detailed Related Work

**Enhancing Diversity in T2I Diffusion Models.** T2I diffusion models are capable of generating high-fidelity and realistic images (Ho et al., 2020; Rombach et al., 2022; Dhariwal and Nichol, 2021). However, the lack of diversity in T2I diffusion models has received considerable attention in recent years (Wan et al., 2024a; Tang et al., 2024). Diversity in T2I generation can be broadly divided into two categories: 1) *demographic diversity*, which aims to mitigate societal biases in T2I models and ensure they generate images that represent diverse demographic groups (*e.g.*, sex, skin tone, and age) (Wan et al., 2024a); and 2) *sample diversity*, which seeks to promote the reusability of prompts to avoid repetitive or overly similar outputs (Tang et al., 2024) and ensure the generated samples can capture the full variability of real-world samples (Kynkäänniemi et al., 2019; Naeem et al., 2020). To enhance demographic diversity, several approaches have been proposed (Bansal et al., 2022; Friedrich et al., 2023; Shrestha et al., 2024; Esposito et al., 2023; He et al., 2024). A particularly straightforward method is to directly augment the original user prompt. For instance, Ethical Intervention (Interven) (Bansal et al., 2022) generalizes the prompt by appending phrases such as "from diverse cultures" or "irrespective of their gender". In addition, Text Augmentation (TextAug) (Ding et al., 2021) enhances the

prompt by explicitly specifying demographic groups with balanced probability, such as adding "This person is a female" to the original prompt. Besides, FairDiff (Friedrich et al., 2023) employs fair guidance to ensure the generated images exhibit the desired demographic attributes. Moreover, DAL (Shen et al., 2024) introduces a distributional alignment loss to direct the generated images toward a user-defined fairness distribution. Additionally, ITI-GEN (Zhang et al., 2023a) utilizes the reference images to learn an inclusive prompt for different demographic attributes. To improve ITI-GEN, FairQueue (Teo et al., 2024) applies the original prompt during the initial timesteps and the ITI-GEN learned prompts in the later stages of the denoising process. Recently, building on Retrieval-Augmented Generation (RAG) (Lewis et al., 2020; Cai et al., 2022), FairRAG (Shrestha et al., 2024) uses the user prompt (*e.g.*, "Photo of a doctor") to retrieve relevant images from external databases and increases the sampling frequency of reference images from minority demographic groups to introduce fairness. Compared to the numerous works on enhancing demographic diversity (Luo et al., 2024; Chuang et al., 2023; Fraser et al., 2023; Gandikota et al., 2024), there is a lack of methods for improving sample diversity for diffusion-based models (*e.g.*, color diversity (Zameshina et al., 2023; Li et al., 2024a; Rassin et al., 2024)). Furthermore, it has become increasingly evident that the state-of-the-art (SOTA) T2I models suffer from insufficient sample diversity (Marwood et al., 2023; Cao et al., 2024). For instance, Tang *et al.* (Tang et al., 2024) demonstrated that for the same prompt, after 50-200 random seeds, the images generated by Stable Diffusion XL (Podell et al., 2023) or DALL-E 3 (Betker et al., 2023) begin to resemble previously generated images. This finding also aligns with our empirical observation in Tab. 1 that the generated samples struggle to capture the variability of real samples. To bridge this gap, we propose a method that enhances both demographic and sample diversity while preserving sample fidelity.

**Guidance in Diffusion Models.** Diffusion models use guidance (Dhariwal and Nichol, 2021; Ho and Salimans, 2022; Brack et al., 2023; Friedrich et al., 2023; Nichol et al., 2021; Bansal et al., 2023; Epstein et al., 2023) to steer the sampling process to ensure that the generated content aligns more closely with the conditioning information. Most notably, Classifier Guidance (CG) (Dhariwal and Nichol, 2021) explicitly incorporates class information into the sampling process by combining the original score estimate with the gradient of the log-likelihood from an auxiliary classifier model. The effect of CG is that data points with a higher probability of being correctly predicted are more likely to be sampled. Classifier-Free Guidance (CFG) (Ho and Salimans, 2022) mathematically interprets that the gradient of the classifier can be expressed using an unconditional score estimate and a conditional score estimate, thereby eliminating the need for an explicit classifier, and empirically verifies that CFG can achieve the same effect as CG. On the basis of CFG, Semantic Guidance (SEGA) (Brack et al., 2023) adds a new term that introduces additional semantic information (*e.g.*, female and male) from text to align the generated images with specific concepts. Unlike SEGA, the proposed guidance method, Bimodal Classifier-Free Guidance (BCFG), incorporates additional information from images, which enhances sample diversity as images offer more detailed information than text. In contrast to CG and CFG, which may trade off sample diversity and fidelity, BCFG improves sample diversity while preserving sample fidelity, as empirically studied in Fig. 14. We hypothesize that this is achieved through the introduction of the image guidance scale, which enables the model to independently control the influence of the image modality and effectively leverages the additional diversity provided by the reference images.

**Adapter in Diffusion Models.** Recently, the concept of adapters, which originated in the NLP community (Houlsby et al., 2019; Zhang et al., 2023c; Li et al., 2023d; Zeng et al., 2024), has been adapted to enable the pre-trained diffusion models for conditioning on new modalities (*e.g.*, image) beyond the input text prompt in a lightweight manner (Mou et al., 2024; Zhang et al., 2023b; Voynov et al., 2023; Smith et al., 2023; Lin et al., 2024a; Kumari et al., 2023). Most notably, Uni-controlnet (Zhao et al., 2024) and Fair-RAG (Shrestha et al., 2024) concatenate the original CLIP text embeddings (Radford et al., 2021) with the image embeddings which are extracted by CLIP followed by a trainable linear projector. However, concatenation may not be an effective method to embed image features into the pre-trained model (Ye et al., 2023). Instead of using concatenation, IP-Adapter (Ye et al., 2023) fuses text and image information by fine-tuning the weight matrices related to the image modality in the cross-attention layers of the U-Net within diffusion models. Furthermore, based on our empirical study in Appendix 4.2, the existing adapters (Ye et al., 2023; Mou et al., 2024), which fully fine-tune the parameters with respect to (w.r.t.) the image modality while freezing the parameters w.r.t. the text modality, may suffer from catastrophic forgetting (Smith et al., 2023) (*e.g.*, the generated images are dominated by image conditioning and lose awareness of text conditioning).

Thus, we propose a low-rank image adapter to balance between ineffectiveness and overly strong image conditioning, aligning better with our objective of introducing additional variation to the pre-trained T2I models.