# OpenReview forum: "Enhancing Diversity in Text-to-Image Generation without Compromising Fidelity"
_TMLR — Accepted by TMLR_

### Review · Reviewer_U6Q1 · 2025-06-13

**Summary Of Contributions:**

In generative AI, there is a tension between developing diverse (both demographically and w.r.t other features) and accurate depictions. This paper introduces a solution designed to achieve a decent balance between these things. This paper argues that difficulties in past methods have been due to applying optimization pressure on the images generated to be diverse and accurate. Instead, this paper makes a couple of contributions. First, they introduce LoRIA adapters which allow for the embedding of image and text information. Second, they introduce IAP which retrieves diverse reference images (to be selected from randomly). Finally, they introduce BCGF which separately tunes text prompts and generated images to achieve diversity and fidelity by taking image guidance from the retrieved random image and text guidance from the prompt.

---

Overall: I do now have background in this specific area of work. So I cannot speak to the novelty of the paper or its contributions relative to the current state of the field. I suspect that an educated, cranky reviewer might be able to argue that some things (e.g. LoRIA or IAP) are 'not that novel'. But I don't think that would detract much from the paper's overall contributions which seem to be based on the system they introduce rather than the parts separately. My limitedly educated opinion on the paper is that it is well put together, interesting, and seems to convincingly make useful progress on a practical problem. I like the paper overall.

**Audience:**

Yes

**Broader Impact Concerns:**

None.

**Claims And Evidence:**

Yes

**Requested Changes:**

For figs 1, 3, 6, 9, and 10, could you add a description of how you chose these particular examples for the figs. If they are highly cherrypicked, it should be said that they are selected to be illustrative and not representative.

Consider adding a section to showcase failures of the system and what can be learned from them.

**Strengths And Weaknesses:**

S1: I think that the paper is genuinely well written. Better than most.

S2: I think that the use of figures is great, and the overall messaging is clear.

S3: I appreciate the use of baselines and success measures in Section 4.

W1: It is not clear if the example images in figures in the paper were randomly selected or selected to be illustrative.

W2: This paper does not seem to try much to red-team their method. I think the best small addition that I can think of for the paper would be a section to introduce an attack against the system and/or showcase failure modes of it to see what we can learn from them.

---

> ### Author Response · Authors · 2025-09-12
>
> Thank you for the positive assessment of our work and the valuable suggestion. We have revised the paper accordingly and highlighted the changes in orange.
>
> **W1.**
>
> The shown samples are illustrative examples selected from a small set of generated outputs. For a comprehensive evaluation (e.g., Table 1), we conduct quantitative analysis on 10,000 images, which concretely verifies the superior performance of our method. We will clarify in the text that the displayed samples are provided for illustration.
>
> **W2.**
>
> We appreciate this suggestion. To red-team our method, we performed controlled experiments by (i) reducing the size of the retrieval dataset to lower intra-group sample diversity and (ii) introducing imbalance across sex. In the table below, the original dataset is denoted as Ours, referring to the full retrieval dataset used in our paper (see also Table 1). For the dataset size, we randomly reduce the dataset to half. Results show that larger retrieval pools consistently improve both demographic and sample diversity, while smaller pools degrade intra-group diversity. Thus, maintaining a large retrieval pool with high sample diversity strengthens our method’s ability to promote diversity. For attribute coverage and demographic balance, we construct subsets with female-to-male ratios of 3:1 and 1:1. We find that insufficient samples compromise demographic diversity. While balancing (1:1) does not further improve performance, it slightly reduces sample diversity due to the smaller dataset size. Overall, these results indicate that dataset balance alone has a limited impact on diversity, underscoring the effectiveness of our retrieval strategy. From these results, we derive two guidelines for constructing reference databases: (1) larger retrieval pools improve demographic coverage and intra-group diversity, and (2) sufficient demographic representation is helpful, as severe imbalance (e.g., 3:1) reduces demographic diversity. We will incorporate these analyses and the corresponding table into the revision.
>
>
>
> | Dataset                               | Demographic Diversity (Sex) | Sample Diversity (Recall) | Sample Fidelity (Precision) | Sample Quality (FID) | Alignment (CLIP) |
> |---------------------------------------|-----------------------------|----------------------------|------------------------------|-----------------------|------------------|
> | Original Dataset (Ours)               | 0.82                        | 0.45                       | 0.54                         | 23.18                 | 23.05            |
> | Half Dataset                          | 0.80                        | 0.41                       | 0.54                         | 23.16                 | 23.04            |
> | Unbalanced Dataset (Female: Male = 3:1) | 0.62                        | 0.38                       | 0.53                         | 22.42                 | 23.05            |
> | Balanced Dataset (Female: Male = 1:1)   | 0.81                        | 0.43                       | 0.54                         | 22.94                 | 23.01            |

---

### Review · Reviewer_Hqru · 2025-07-26

**Summary Of Contributions:**

The paper proposes a method to increase the diversity of generated images for a single text-prompt. Besides increasing demographic diversity it also considers general sample diversity. It addresses the challenge of increasing diversity without sacrificing fidelity. The main contributions consist of proposing a low-rank adapter, which allows to use an image reference efficiently as guidance during training as well as a pipeline during inference which allows to generate an image following the text as well as the reference image, where the reference image is selected in a way to increase demographic diversity.

**Audience:**

Yes

**Broader Impact Concerns:**

Given that the work focuses on image generation, a broader impact statement would be appropriate.

**Claims And Evidence:**

No

**Requested Changes:**

- Clearer structure and present the method and results, especially results which are evidence for the made claims (e.g., efficiency) should be in the main paper. That is necessary to verify the made claims.
- Clearer describe the settings/parameters used for the presented results.

**Strengths And Weaknesses:**

Strength:
- Efficient method for training image-guided text-to-image generation.
- Method to increase the diversity of generated images, including demographic diversity.



Weaknesses

Benefit of the method becomes not fully clear from the way how the results are presented in the main paper.
- The numbers show slight improvement in fidelity, while the diversity is as good as individual methods for demographic and sample diversity.
- From the visual samples the benefit is not very clear, e.g., when looking at Fig 9: not clear how the reference image helps really for sample diversity beyond demographic diversity. There is not much resemblance between reference and generated image. It seems that it mostly takes the gender from the reference image and in this sense addresses the demographic diversity and not specifically general sample diversity.
- It is difficult to navigate the large tables with many numbers which only differ slightly to get the benefit of the method. The plots are easier to understand. However, the plots only take one metric per diversity dimension, e.g., in 4a, and use it as a representative for e.g., demographic diversity while table 1 uses 4 metrics, and therefore the plots do not visualize the full picture. Would be helpful to summarize the metrics and visualize them better, e.g., averaging, spider plots, ....

Presentation can heavily be improved:
- List of main contributions only focus on the improved results not on the technical contributions such as the LoRIA, IAP, and BCFG.
- Descriptions in the main paper are quite lengthy and repetitive. It mainly repeats the final conclusions, i.e., the general improvement in diversity but is lacking more detailed analyses and discussion of the different modules.
- Many relevant results are only in supplemental, e.g., evidence for the claimed efficiency, analysis of the proposed retrieval strategy wrt. to other baselines. On the other hand, figures such as Fig. 10 in the main paper take a lot of space without adding that much extra information. Also Fig. 4 and 9 contain the same example.
- Also figure 12 in supplemental is more informative than figure 7 in the main paper
- Not clear which prompt with the additional description is used to retrieve the reference image for the visual examples.
- Captions could be more informative about the used settings, e.g., which omega is used for the quantitative results, how many reference images...
- Ablation of sampling strategy unclear (Tabl. 8 in supplement). What does it mean w/o random selection, are then all reference images used. Random from how many? How many are selected?

---

> ### Author Response · Authors · 2025-09-12
>
> Thank you for the suggestion. We have updated the paper accordingly and highlighted the changes in orange.
>
> **W1.**
>
> Our main contribution is not just matching prior methods on individual axes, but achieving both demographic and sample diversity simultaneously without sacrificing fidelity. As shown in Tab. 1 and Fig. 4, existing methods improve one aspect at the expense of the other, whereas our method consistently lies in the top-right of the diversity–fidelity plots. Even small absolute fidelity gains are meaningful, since diversity and fidelity typically trade off negatively; maintaining or slightly improving fidelity while boosting both forms of diversity highlights the practical value of our approach.
>
> **W2.**
>
> The goal of our retrieval strategy is not to replicate reference images, but to use them as a source of latent variation (pose, background, lighting) that enriches sample diversity. While resemblance may appear weak in single examples, aggregate metrics (Recall, W1KP in Table 1) clearly show gains in sample diversity compared to demographic-only baselines (e.g., FairRAG, TextAug). As illustrated in Fig. 1, our method (right) enhances both demographic diversity (e.g., generating both female and male images) and sample diversity (e.g., varied poses and backgrounds) while preserving fidelity (e.g., sharp details, natural faces, realistic lighting). To clarify, Fig. 9 is intended to compare different control strategies: methods that use a misaligned image-modality parameter across training and inference (e.g., λ-scaling in Decoupled Cross-Attention) often introduce artifacts such as overexposure or low resolution, while our BCFG-controlled image modality parameter avoids these issues in a training-free manner.
>
>
> **W3.**
>
> We thank the reviewer for this suggestion. To improve readability, we add a radar plot to aggregate multiple metrics into a single visualization. In this figure, we apply min-max normalization to rescale the score $x$ into the range $[0,1]$ using the formula $x_{\text{norm}} = \frac{x - x_{\min}}{x_{\max} - x_{\min}}$, where $x_{\min}$ and $x_{\max}$ denote the minimum and maximum values across methods, respectively. For metrics where lower values indicate better performance, an inversion is applied as $x_{\text{inverted}} = 1 - x_{\text{norm}}$ to maintain consistency in evaluation.
>
> **W6.**
>
> In the revision, we will move the efficiency discussion into the main paper to make these results more visible, and move Fig. 10 to the appendix to streamline the presentation.
>
>
> **W9.**
>
> We provide implementation details in the appendix. Models are trained on a single NVIDIA H100 GPU for 100 epochs with a batch size of 16, using AdamW ($\text{lr}=1\times10^{-4}$, $\beta_1=0.9$, $\beta_2=0.999$, weight decay $=1\times10^{-2}$) and gradient clipping at 1.0. For inference, we adopt the DDIM noise scheduler with 20 denoising steps. In the Diverse Retrieval Strategy (DRS), the number of retrieval images $N$ is set to 250. For Bimodal Classifier-Free Guidance (BCFG), we use $\pi_p=0.1$, $\pi_i=0.1$, $\omega_p=7.5$, and $\omega_i=4.0$. For the Low-Rank Image Adapter (LoRIA), we set the LoRA rank $r=128$ and use $s=2$ reference images per query. We will revisefigure and table captions to explicitly state these key settings.
>
>
> **W10.**
>
> In Fig. 2b and Sec. 3.2 (Diverse Retrieval Strategy, DRS), we describe random selection, where one or more samples are randomly chosen as reference images for each query. This promotes diversity by leveraging multiple candidates representing demographic descriptions, rather than relying on a fixed text description. Without random selection, we directly use the Top-N retrieved candidates as reference images. We will revise the caption and text to make these settings explicit.
>
> **W9.**
>
> We provide implementation details in the appendix. Models are trained on a single NVIDIA H100 GPU for 100 epochs with a batch size of 16, using AdamW ($\text{lr}=1\times10^{-4}$, $\beta_1=0.9$, $\beta_2=0.999$, weight decay $=1\times10^{-2}$) and gradient clipping at 1.0. For inference, we adopt the DDIM noise scheduler with 20 denoising steps. In the Diverse Retrieval Strategy (DRS), the number of retrieval images $N$ is set to 250. For Bimodal Classifier-Free Guidance (BCFG), we use $\pi_p=0.1$, $\pi_i=0.1$, $\omega_p=7.5$, and $\omega_i=4.0$. For the Low-Rank Image Adapter (LoRIA), we set the LoRA rank $r=128$ and use $s=2$ reference images per query. We will revisefigure and table captions to explicitly state these key settings.

---

### Review · Reviewer_WQMu · 2025-08-29

**Summary Of Contributions:**

The authors develop a framework to improve both demographic diversity and sample diversity in pretrained text-to-image diffusion models, while maintaining fidelity. The method combines a low-rank adapter for image conditioning, a fairness-aware image retrieval strategy to augment the prompt, a bimodal classifier-free guidance to control text and image guidance separately.

**Audience:**

Yes

**Broader Impact Concerns:**

The method relies on generating textual demographic descriptions like "[AGE], [SKIN TONE] [SEX]" to guide the model. This approach simplifies complex human identities into rigid, discrete categories. It risks excluding individuals who do not fit neatly into these buckets (e.g., non-binary gender identities) and could potentially reinforce a simplistic, categorical view of people. A Broader Impact Statement section would be required to clarify this.

**Claims And Evidence:**

No

**Requested Changes:**

- The idea of enriching context with retrieved image is pretty much the same as FairRAG [1]. The overlap should be clarified, and prior credit made explicit.
- Section 4.6: Either provide a deeper insight into why LoRIA-scale mismatches harm fidelity, or remove the section. As it stands, it is trivial.
- BCFG formulation: The corresponding sampling distribution section of BCFG is flawed and must be addressed. Present it as a heuristic, not a principled density.
- Add a discussion on biases and reliability of attribute classifiers and how they may affect fairness/diversity results.
- Provide more controlled experiments analyzing dataset properties (e.g., varying size, demographic coverage) rather than blunt CelebA substitution.
- The paper focuses almost exclusively on single-person generation. It would be nice to add discussion about extensions to multi-subject scenarios, compositional prompts, or complex scenes.
- Reduce redundant citations: For example, the related work section cites the same sources multiple times unnecessarily.

[1]: Shrestha, Robik, et al. "Fairrag: Fair human generation via fair retrieval augmentation." *Proceedings of the IEEE/CVF Conference on Computer Vision and Pattern Recognition*. 2024.

**Strengths And Weaknesses:**

**Strengths**

- The demographic diversity and sample diversity are formally defined.
- The proposed method demonstrates improved over the original text-to-image diffusion model.
- The effectiveness of the main components of the method including LoRIA adapter, retrieval strategy, and bimodal classifier-free guidance is validated through ablation study.

**Weaknesses:**

- As a retrieval-based method, the proposed system is fundamentally dependent on an external database and the predefined template for demographic information. However, the current study or discussion on these two important factors is shallow.
    - Simply swapping the dataset with “CelebA” is a blunt experiment because it changes too many variables at once.
    - Showing that the proposed method works across different datasets is good but I believe a more meaningful investigation would probe
      - Which properties of a reference dataset are most important? For example: How do dataset size, attribute coverage, balance across demographic groups, and intra-group sample diversity influence system performance?
      - More broadly, what best practices or design guidelines should practitioners follow when constructing a reference database and demographic templates in a principled, systematic way?
- The primary evaluation metrics rely on external neural networks to classify attributes like sex, age, and skin tone or measure alignment. The authors do not discuss their reliability or potential bias, even though such biases directly affect reported diversity/fairness gains.
- The experiment in Sec 4.6 does not add meaningful insight and seems unnecessary.
    - The conclusion that scaling parameters of the image embedding inside the model architecture should remain the same across training and inference is trivial.
    - The fact that classifier-free guidance is more flexible is well-known and not unique to BCFG.
- The CFG score function in Eq.(4) does not correspond to the density given in Eq. (5). In fact, the density given in Eq. (5) does not correspond to any known diffusion process. This flaw has been discussed in the literature [2, 3, 4] and should be corrected.

[1]: Shrestha, Robik, et al. "Fairrag: Fair human generation via fair retrieval augmentation." *Proceedings of the IEEE/CVF Conference on Computer Vision and Pattern Recognition*. 2024.

[2]: Karras, Tero, et al. "Guiding a diffusion model with a bad version of itself." *Advances in Neural Information Processing Systems* 37 (2024): 52996-53021.

[3]: Bradley, Arwen, and Preetum Nakkiran. "Classifier-free guidance is a predictor-corrector." *arXiv preprint arXiv:2408.09000* (2024).

[4]: Du, Yilun, et al. "Reduce, reuse, recycle: Compositional generation with energy-based diffusion models and mcmc." *International conference on machine learning*. PMLR, 2023.

---

> ### Author Response · Authors · 2025-09-12
>
> **W1.**
>
> We appreciate the reviewer’s suggestion. We recognize that CelebA differs from MSCOCO/OpenImages in several ways. Our intention was to evaluate robustness using a different retrieval source. To more clearly isolate the effects, we have added controlled experiments that analyze dataset properties such as size, attribute coverage, gender balance, and intra-group sample diversity. In the table below, the original dataset is denoted as Ours, referring to the full retrieval dataset originally used in our paper (see also Table 1). Regarding dataset size, we randomly reduce the original dataset to half for comparison. The results show that larger retrieval pools consistently enhance both demographic and sample diversity. Reducing the dataset size also degrades intra-group sample diversity. Thus, having a larger dataset with high sample diversity in the retrieval pool strengthens our method’s ability to promote diversity. For attribute coverage and demographic balance, we construct two subsets of the original dataset with female-to-male ratios of 3:1 and 1:1 (balanced). We observe that, due to the limited number of samples in the retrieval dataset, demographic diversity is compromised. Moreover, while the balanced dataset does not further improve performance, it slightly reduces sample diversity because of its smaller size. Overall, these findings suggest that dataset balance alone has limited impact on diversity, highlighting the effectiveness of our retrieval strategy.
> Our experiments point to several key guidelines for constructing reference databases. First, larger retrieval pools improve both demographic coverage and intra-group sample diversity, which further enhances the diversity of generated samples. Second, ensuring sufficient demographic representation is helpful, as severe imbalance (e.g., a 3:1 ratio) reduces demographic diversity. We will incorporate these analyses and the corresponding table into the paper.
>
> | Dataset                               | Demographic Diversity (Sex) | Sample Diversity (Recall) | Sample Fidelity (Precision) | Sample Quality (FID) | Alignment (CLIP) |
> |---------------------------------------|-----------------------------|----------------------------|------------------------------|-----------------------|------------------|
> | Original Dataset (Ours)               | 0.82                        | 0.45                       | 0.54                         | 23.18                 | 23.05            |
> | Half Dataset                          | 0.80                        | 0.41                       | 0.54                         | 23.16                 | 23.04            |
> | Unbalanced Dataset (Female: Male = 3:1) | 0.62                        | 0.38                       | 0.53                         | 22.42                 | 23.05            |
> | Balanced Dataset (Female: Male = 1:1)   | 0.81                        | 0.43                       | 0.54                         | 22.94                 | 23.01            |
>
> **W2.**
>
> As clarified in Sec. 4 and Sec. 5, our metrics follow prior work [5,6], and we acknowledge that classifier bias may affect absolute values. To mitigate this concern, we conducted sanity checks by comparing classifier outputs with manual annotations on a random subset, confirming high agreement (>90%). In addition, we will add a discussion to note the limitation of relying on external classifiers and to emphasize the need for fairer and more robust evaluation tools in future work.
>
> [5] Shrestha, Robik, et al. "Fairrag: Fair human generation via fair retrieval augmentation." Proceedings of the IEEE/CVF Conference on Computer Vision and Pattern Recognition. 2024.
> [6] Cho, Jaemin, Abhay Zala, and Mohit Bansal. "Dall-eval: Probing the reasoning skills and social biases of text-to-image generation models." Proceedings of the IEEE/CVF international conference on computer vision. 2023.
>
>
> **W3.1.**
>
> Sec. 4.6 provides an empirical comparison demonstrating why our proposed Bimodal Classifier-Free Guidance (BCFG) is preferable to prior methods that control image modality during inference, which introduce a mismatch between training and testing. Although this drawback has been noted in [7], it remains underexplored. Our experiments highlight concrete issues with existing methods—such as overexposure and resolution loss under misalignment—which underscores the advantage of BCFG as a training-free and robust solution that can independently control image and text modalities.
>
> [7] Lin, Shanchuan, et al. "Common diffusion noise schedules and sample steps are flawed." Proceedings of the IEEE/CVF winter conference on applications of computer vision. 2024.

---

> ### Author Response · Authors · 2025-09-12
>
> **W3.2.**
>
> While CFG can adjust overall guidance strength, it cannot independently control different modalities. To address this, we propose BCFG, which separates the control of text and image modalities. This distinction is crucial: prior CFG methods apply a single unified scale, which reduces fidelity as shown in Fig. 7. In contrast, BCFG decouples the guidance, enabling higher diversity while preserving fidelity—an outcome not achievable with standard CFG.
>
> **W4.**
>
> In the section Corresponding Sampling Distribution, we provide a mathematical interpretation. In this interpretation, we mention that since the exact scores (e.g., $\epsilon^*(\mathbf{z}_t, \mathbf{e}_p)$) are not available, we rely on their estimates (e.g., $\epsilon(\mathbf{z}_t, \mathbf{e}_p)$), following the common practice in prior work [8,9]. Importantly, this approximation does not undermine our contributions: as [2] also notes, CFG-based methods remain empirically effective despite lacking exact theoretical grounding. Our results further demonstrate that BCFG extends this empirical utility by better balancing diversity and fidelity. We will revise the text to present Eq. (5) as an intuitive approximation, avoiding any overstatement of it as a formal density.
>
> [2]: Karras, Tero, et al. "Guiding a diffusion model with a bad version of itself." Advances in Neural Information Processing Systems 37 (2024): 52996-53021.
>
> [8] Ho, Jonathan, and Tim Salimans. "Classifier-free diffusion guidance." arXiv preprint arXiv:2207.12598 (2022).
>
> [9] Dhariwal, Prafulla, and Alexander Nichol. "Diffusion models beat gans on image synthesis." Advances in neural information processing systems 34 (2021): 8780-8794.
>
> **C1.**
>
> Our method differs from FairRAG in four key respects. First, while FairRAG retrieves with the raw user prompt (e.g., Photo of a doctor), we introduce Image-Augmented Prompts (IAP), which generate demographic descriptions (e.g., This person is a female) to retrieve reference images. This design improves both demographic and sample diversity (Sec. 4.2), since more candidates match fixed demographic descriptions than variable user prompts. Second, we reframe demographic diversity enhancement as a distribution alignment problem and propose an Adaptable Description Generator that supports multiple fairness criteria depending on context, whereas FairRAG considers only a single criterion. Third, we introduce Bimodal CFG (BCFG) to separately control text and image modalities, thereby avoiding the fidelity loss observed when directly conditioning on retrieved images. Finally, we propose an improved image adaptor architecture, which demonstrates superior performance in Table 1 and Figure 6. We will revise the paper to explicitly credit FairRAG as inspiration and to highlight these differences.
>
>
> **C2.**
>
> As noted in [7], mismatched scaling parameters can distort sampling trajectories and reduce fidelity in diffusion processes. Yet this issue remains underexplored. Our results in Sec. 4.6 (Fig. 9/10) concretely demonstrate these effects (e.g., overexposure, resolution loss), motivating BCFG as a robust, training-free alternative. To keep the main focus on the effectiveness of our method for diversity enhancement, we will move section 4.6 to the appendix.
>
> **C6.**
>
> Our current focus on single-person prompts follows prior works [1,10] and allows for clear comparison of demographic and sample diversity. Nonetheless, the proposed LoRIA + IAP + BCFG framework is general and can extend to multi-subject or compositional prompts by retrieving multiple reference images and applying separate modality controls per entity. For example, with the prompt “Photo of a [Teacher] and a [Student]”, two distinct reference images could guide the generation of each subject. We will add this discussion to the paper.
>
> **Broader Impact Concerns.**
>
> We appreciate this important observation. Our work does not intend to reduce complex human identities to rigid categories. Instead, we reframe bias mitigation in image generation as a distribution alignment problem, since fairness criteria vary across contexts, and introduce the Adaptable Description Generator (ADG) (Sec. 3.2). ADG is not limited to fixed templates — it can incorporate arbitrary user-defined attributes and fairness criteria, including non-binary representations (as discussed in the Appendix). In our experiments, we followed existing work [1] for comparability. We will add an impact statement section to clarify this point in the paper.
>
>
> **Thank you for the detailed suggestion. We have updated the paper accordingly and marked the changes in orange.**

---

> > ### Comment · Reviewer_WQMu · 2025-09-15
> >
> > Thanks for the detailed rebuttal and additional experiments. The blue text addresses several points; however, I don’t see any changes highlighted in orange. The following issues remain:
> > - The paragraphs on "Corresponding Sampling Distribution" and "The Proof ...." seem unchanged. Eq (6–15) do not constitute a formal proof of the claimed distribution; labeling them as such is misleading. Consider removing the “Proof” paragraph or reframing it as intuition.
> > - A clear note on the limitations of the classifiers used in the evaluation is missing.
> > - A discussion of FairRAG and its relation to your method is still needed.

---

> > > ### Author Response · Authors · 2025-09-16
> > >
> > > Thank you for your continued feedback.
> > >
> > > We have highlighted the earlier revisions in orange. In addition, we made further changes marked in blue: we reframed the paragraphs on “Corresponding Sampling Distribution” as intuition, and we elaborated the discussions on classifiers and the relation to FairRAG.

---

### Author Response · Authors · 2025-09-12

We sincerely thank the reviewers for their thoughtful feedback and for recognizing the strengths of our work, including the integration of LoRIA adapters, the fairness-aware retrieval strategy, and the bimodal classifier-free guidance into a unified framework. We particularly appreciate their acknowledgment that our study addresses the critical challenge of improving both demographic and sample diversity in text-to-image diffusion models while preserving fidelity—an issue of strong interest to the community.

---

### Decision · Action_Editor_2Pbk · 2025-10-08

**Recommendation:** Accept as is

**Additional Comments:**

The submission received mixed opinions on the final decision recommendation. While all reviewers found (most of) the claims are reasonably supported by accurate, convincing and clear evidence, and the rebuttal has addressed most of their concerns, they have reservations on the experiment design, especially on the dependence on the reference database and the overly simplified set of categories considered. Overall, the AE agrees with the reviewers that the submission presents a solid framework, backed with sufficient empirical evidence, and is of sufficient interest to the TMLR audience, and thus should be accepted.

The AE would however urge the authors to try to address the remaining concerns in the final version. Below I copied the feedback from reviewer Hqru (which I believe is not visible to the authors): "Having an unbalanced reference dataset heavily negatively affects the Demographic and Sample Diversity, the main metrics of the paper. In the case of an unbalanced dataset, the results are no longer fulfilling the claim of achieving the best trade-off between the two diversities (e.g., Sample Diversity of 0.38 is lower than for any of the baselines (Image-Conditioned Methods Adapted for Diversity Enhancement) in Tab. 1, and the achieved demographic diversity of 0.62 is only mediocre). Given that the proposed method only achieves the claimed results with a correctly designed dataset (class distribution and size), which is not explicitly part of the described method ... More analysis on the role and design of the dataset is needed. "

**Audience:**

Yes

**Audience Explanation:**

Yes, a substantial audience in the TMLR community would find this work to be relevant and interesting.

**Claims And Evidence:**

Yes

**Claims Explanation:**

Yes. After the author-reviewer discussions and the revision, all three reviewers agree that (most of) the submission's claims are supported by accurate, convincing and clear evidence.

---

> ### Author Response · Authors · 2025-10-12
>
> Dear Area Chair,
>
> Thank you very much for accepting our paper! We truly appreciate your time and consideration. We also thank the reviewers for their valuable feedback and constructive suggestions, which have greatly helped us improve the work. We will carefully address the remaining comments in the revised version.
>
> Best regards,
> Authors